# Dual protein kinase and nucleoside kinase modulators for rationally designed polypharmacology

Kahina Hammam[1], Magali Saez-Ayala[1], Etienne Rebuffet[1,2], Laurent Gros[2], Sophie Lopez[1], Berengere Hajem[2], Martine Humbert[2], Emilie Baudelet[1], Stephane Audebert [1], Stephane Betzi[1], Adrien Lugari[1], Sebastien Combes[1], Sebastien Letard[1], Nathalie Casteran[2], Colin Mansfield[2], Alain Moussy[2], Paulo De Sepulveda[1], Xavier Morelli [1] & Patrice Dubreuil [1]

Masitinib, a highly selective protein kinase inhibitor, can sensitise gemcitabine-refractory cancer cell lines when used in combination with gemcitabine. Here we report a reverse proteomic approach that identifies the target responsible for this sensitisation: the deoxycytidine kinase (dCK). Masitinib, as well as other protein kinase inhibitors, such as imatinib, interact with dCK and provoke an unforeseen conformational-dependent activation of this nucleoside kinase, modulating phosphorylation of nucleoside analogue drugs. This phenomenon leads to an increase of prodrug phosphorylation of most of the chemotherapeutic drugs activated by this nucleoside kinase. The unforeseen dual activity of protein kinase inhibition/nucleoside kinase activation could be of great therapeutic benefit, through either reducing toxicity of therapeutic agents by maintaining effectiveness at lower doses or by counteracting drug resistance initiated via down modulation of dCK target.

[1] Centre de Recherche en Cancérologie de Marseille (CRCM), INSERM, CNRS, Aix-Marseille Univ, Institut Paoli-Calmettes, Equipe Labellisée Ligue, Marseille 13009, France. [2] AB Science, Paris 75008, France. Kahina Hammam, Magali Saez-Ayala, Etienne Rebuffet and Laurent Gros contributed equally to the work. Xavier Morelli and Patrice Dubreuil jointly supervised the work. Correspondence and requests for materials should be addressed to X.M. (email: xavier.morelli@inserm.fr) or to P.D. (email: patrice.dubreuil@inserm.fr)

In last 10 years, important progresses have been reported in the field of pharmacologic targets identification and in the development of new bioactive drugs. These progresses were enabled, among other breakthroughs, by technological advances in genomics, proteomics and structural biology[1–3]. The general practice of all these techniques together led to the identification of highly selective and potent drugs to cure cancer, as well as treating other pathologies, such as chronic inflammatory diseases and viral infections. The clinical development and regulatory approval of agents such as Herceptin (trastuzumab)[4], for the treatment of advanced breast cancer, and Gleevec (imatinib)[5], for chronic myelogenous leukaemia and gastrointestinal stromal tumours (GIST), has revolutionised cancer treatment and validated the concept of target-directed therapies. These agents not only prolong life and improve its quality, but they also provide clinical validation of the emerging field of molecular oncology, specifically therapies targeting kinase enzymes that play a critical role in tumorigenesis. Nevertheless, after a few years, these types of treatments are reaching their limits, as drug resistances frequently occur during long-term treatments[6, 7]. Moreover, despite the concept of targeted therapy, this type of chronic treatment often leads to various adverse events. The toxic effects are either due to activity against the main target or against an "unidentified target"[8–10]. This "off target" effect, often deleterious, could also bring important therapeutic benefits.

Masitinib is a selective tyrosine kinase inhibitor (TKI) targeting the c-Kit tyrosine kinase[11], a major pharmacological target in oncology, which is a member of the type III receptor protein-tyrosine kinase family (RTK)[12]. Masitinib has a higher affinity and selectivity in vitro than any other TK inhibitors and does not inhibit multiple kinases, which could be linked to toxic effects. Masitinib also potently inhibits recombinant PDGFR (platelet-derived growth factor receptor) and the intracellular kinases Lyn and Fyn[11]. Additionally, masitinib is active and orally bioavailable, and has

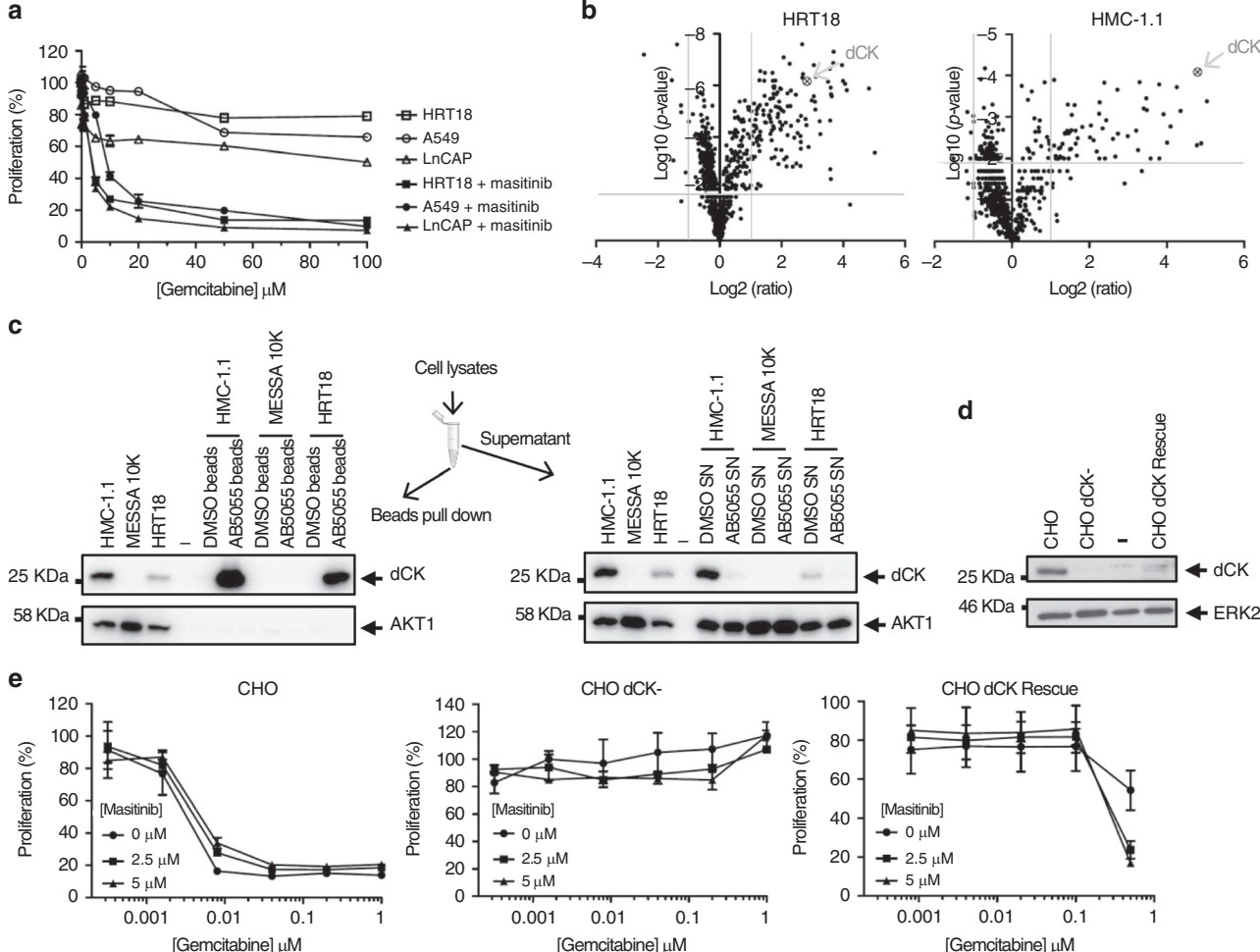

**Fig. 1** Identification of dCK as the target protein responsible for masitinib sensitisation effect on gemcitabine-refractory cancer cells. **a** HRT18 (colon), A549 (lung) and LNCaP (prostate) cancer cell lines were tested in proliferation assays for response to gemcitabine (0–100 μM) in the absence and presence of masitinib (10 μM) (mean ± s.d., n = 3). Addition of masitinib led to a sensitisation of these cancer cell lines to gemcitabine, inducing a shift of the respective IC50 to lower gemcitabine concentrations. **b** Volcano plot illustrating significantly different proteins profiles in HMC-1.1 and HRT18 cell lines. The −log10 (P-value) is plotted against the log2 (fold change of protein abundance in the presence and absence of masitinib). The significance thresholds are represented by a horizontal line (P-value = 0.01) and two vertical lines (twofold change). **c** Visualisation of the specificity of the interaction between dCK and NH₂-modified masitinib (AB5055) on three cell line lysates by pull-down and western blotting (HMC-1.1, MESSA 10K and HRT18). AKT1 antibody was used as negative control. **d** Level of expression of dCK in different stable CHO clones determined by western blot analysis. ERK2 antibody was used as the loading control. **e** Effect of a masitinib/gemcitabine combination treatment on CHO, CHO dCK- and CHO dCK Rescue reconstituted clone (low dCK expression). Cell lines were tested in proliferation assays for response to gemcitabine (0–1 μM) in the presence and absence of masitinib at different concentrations (0, 2.5 and 5 μM). Sensitivity to gemcitabine and to the combination of gemcitabine and masitinib were measured after 5 days of treatment with the drugs (mean ± s.d., n = 3). Reconstitution of dCK in a stable CHO clone restores CHO cell line sensitivity to gemcitabine

been evaluated as a single-agent in phase III clinical trials in pathologies such as mastocytosis[13], mast cell tumour[14] and GIST[15]. Its use in other pathologies as a single agent had no significant anti-proliferative activity, while the combination of masitinib/gemcitabine (a nucleoside analogue) inhibits the growth of human pancreatic adenocarcinoma in vitro and in vivo[16]. A phase III trial of masitinib plus gemcitabine in the treatment of advanced pancreatic cancer has confirmed the survival benefit for patients[17]. This property could not be explained by the kinase selectivity profile of the drug, since masitinib is highly selective towards c-Kit[18]. We thus hypothesised that masitinib specifically targets a protein responsible for this beneficial effect of gemcitabine.

In the present work, we identified deoxycytidine kinase (dCK) as the target responsible for the sensitisation of various cancer cell lines to gemcitabine, using reverse-proteomics. dCK is a key enzyme in the nucleoside salvage pathway and recycles bases and nucleosides originating from the degradation of RNA and DNA to achieve the biosynthesis of deoxyribonucleotides, which are required for DNA replication and repair[19]. This cytosolic nucleoside kinase catalyses the 5′-phosphorylation of 2′-deoxycytosine (2′dC), 2′-deoxyadenosine (2′dA) and 2′-deoxyguanosine (2′dG). dCK is also responsible for the initial phosphorylation of numerous anticancer and antiviral nucleoside analogues. dCK adds the first phosphoryl group to nucleosides analogues and is usually the rate-limiting enzyme of the overall process of converting nucleosides to their deoxynucleoside triphosphate form. Due to its critical role in pro-drug activation, the deficiency of dCK is associated with resistance to nucleoside-like drugs, including gemcitabine[20–22]. Conversely, increasing dCK activity is associated with enhanced activation of these analogues[23–25]. Hence, dCK is clinically important because of its relationship to both drug resistance and sensitivity.

Moreover, we revealed how masitinib, as well as other protein kinase inhibitors, such as imatinib, interact with dCK. This previously unknown molecular interaction leads to unexpected nucleoside kinase activation, which causes an increase in physiological and pro-drug phosphorylation of dCK substrates. In addition, we solved crystal structures of dCK in complex with masitinib and imatinib, and describe how dCK is activated through a conformational-dependent activation mechanism using a dynamic functional pocket.

## Results

**Masitinib sensitises cancer cell lines to gemcitabine.** Masitinib has shown efficacy in pancreatic cancer cell lines[16] and in patients[17], when added to gemcitabine, thus studies were performed on other cancer cell lines to evaluate the in vitro therapeutic potential of masitinib in combination with gemcitabine. Sensitivity to gemcitabine monotherapy and combination treatment with gemcitabine/masitinib were investigated by proliferation assays in prostate (LNCaP), colon (HRT18) and lung (A549) cancer cell lines (Fig. 1a). While gemcitabine alone could not induce strong anti-proliferative effects over a wide concentration ranges, the addition of masitinib led to a sensitisation of these cancer cell lines, inducing a shift of the respective IC50 to lower gemcitabine concentrations, as described previously in human pancreatic adenocarcinoma[16]. In contrast, single-agent masitinib had only a slight anti-proliferative activity in these pathologies (Supplementary Fig. 1), indicating a synergistic interaction for the combined treatment (Supplementary Fig. 2).

**Identification of dCK by reverse proteomic approach.** In order to pinpoint the target responsible for gemcitabine sensitization by masitinib, we conducted a reverse chemoproteomic approach[26] using an NH₂-modified version of masitinib as a probe (Supplementary Fig. 3). Experimental conditions were optimised on

known targets (c-Kit and Lyn) by western blot analysis (Supplementary Fig. 4). Beads cross-linked with masitinib were incubated with cellular lysates of HMC-1.1 cells (human mast leukaemia), a cell line expressing c-Kit and Lyn (masitinib targets), and HRT18 cells (human colorectal adenocarcinoma), a cell line presenting a gemcitabine sensitisation by masitinib. Captured proteins were analysed by LC–MS and identified by protein database comparison. The abundance of proteins relative to the control and statistical significance were calculated. A complete list of the identified proteins and their quantification is enclosed in Supplementary Datas 1–4 and a table showing the on-target and off-target kinases with their abundances is presented in Supplementary Table 1. We identified the masitinib known targets c-Kit and Lyn in HMC-1.1 cellular lysates, a cell line expressing both proteins, and only Lyn in HRT18 cellular lysates, as this cell line does not express c-Kit (Supplementary Table 1). We only identified three other kinases with low abundance, certainly due to the high selectivity profile of masitinib towards c-Kit[18]: the mitochondrial acylglycerol kinase (AGK), the catalytic subunit of the DNA-dependent protein kinase (DNA-PKcs) and the Abelson tyrosine-protein kinase 2 (ABL2). This result is not surprising as these kinases use ATP as phosphate donor and masitinib is an ATP mimic. We also identified the NAD(P)H:quinone oxidoreductase NQO2, which was also found as one of the foremost interactors of other TKIs on the same type of reverse proteomics approach[26]. We rapidly identified deoxycytidine kinase (dCK) among the list of potential candidates because of its direct role in nucleoside pro-drug activation, such as with gemcitabine. This protein was identified with a high score ratio and was listed third in the HMC-1.1 cell line cellular lysates in the fold change relative to control value ranking (Fig. 1b). When using the HRT18 cell line, dCK was ranked twenty-seventh, which supports our previous result, as this cell line has a low dCK expression (Fig. 1c). These data prompted us to evaluate dCK as the protein vector responsible for the masitinib sensitisation effect on gemcitabine-refractory cancer cells.

The specificity of the interaction between masitinib and dCK was next validated in vitro on three cell line lysates (HMC-1.1, HRT18 and Messa 10K (human ovarian sarcoma cell line)) by pull-down and western blotting, using the NH₂-modified masitinib probe (Fig. 1c). HMC-1.1 and Messa 10K cell lysates were taken as positive and negative controls, for their high and null dCK expression, respectively. As shown in Fig. 1c, dCK was detected only in masitinib-coupled beads treated with HMC-1.1 and HRT18 cell lysates, but not in the supernatant. Additionally, AKT1 protein kinase, was not detected in masitinib-coupled beads, indicating that masitinib was able to directly bind dCK in a specific manner.

**Gemcitabine sensitization is related to dCK expression.** To evaluate the applicability of this discovery, we investigated the effect of a masitinib/gemcitabine combination treatment on cell lines expressing different dCK levels. We first confirmed that gemcitabine resistance in numerous cell lines was related to dCK expression (Supplementary Fig. 5a). Western blot analysis showed that dCK was highly expressed in HMC-1.1 cells, a cell line sensitive to gemcitabine treatment, whereas in HRT18 cells, a medium sensitive gemcitabine-cell line, dCK was only moderately expressed. On the other hand, the protein was not expressed in Messa 10K (human ovarian sarcoma), CHO dCK- (chinese hamster ovary dCK deficient cell line) and MiaPaCa2 (pancreatic cancer cell line), three cell lines which are known to be resistant to gemcitabine treatment[27–29]. We then evaluated the effect of masitinib/gemcitabine combination on two of these cell lines (Supplementary Fig. 5b, c). The addition of masitinib to gemcitabine treatment

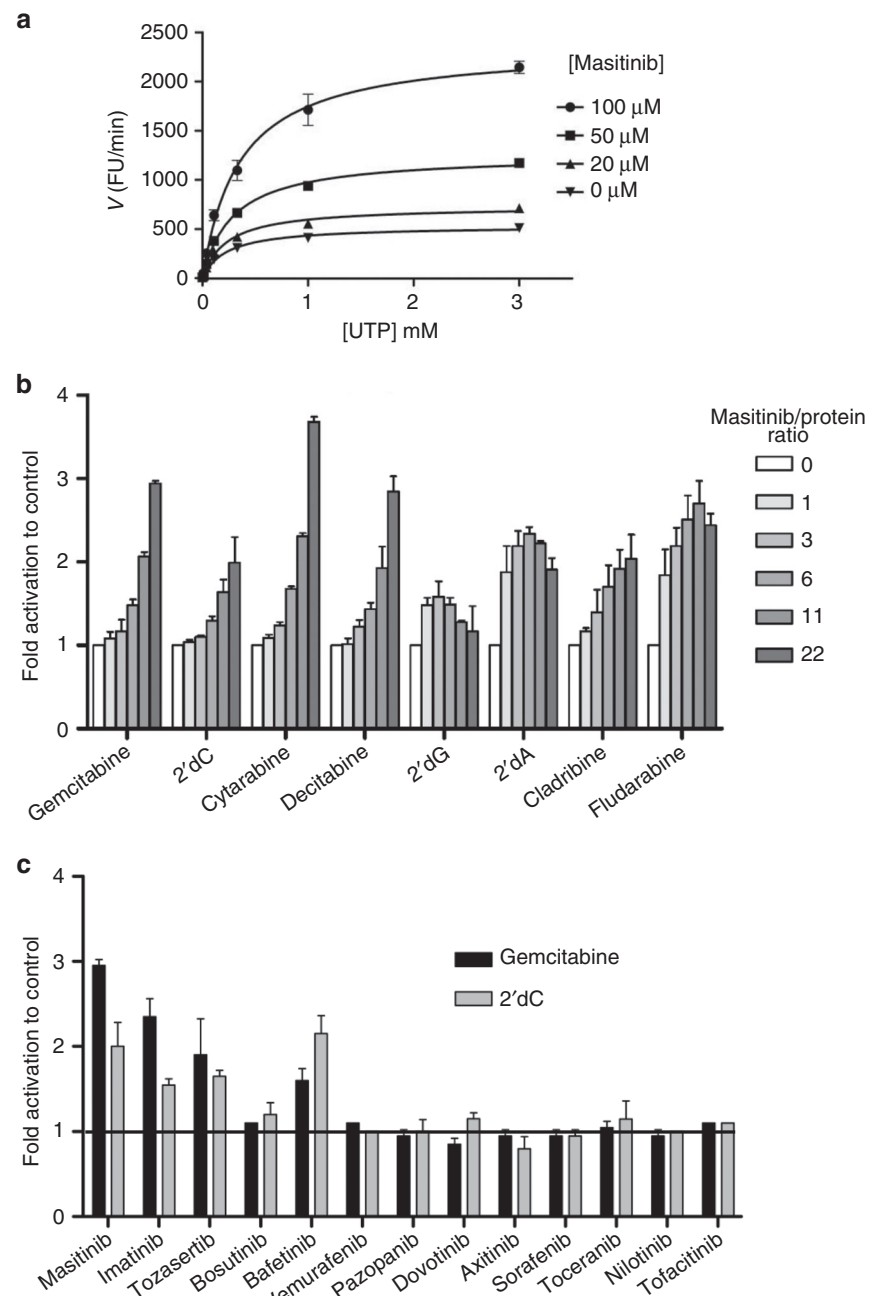

**Fig. 2** Analysis of the effect of masitinib and other kinase inhibitors on substrate phosphorylation by dCK. **a** The effect of masitinib on dCK activity was analysed by varying UTP in the presence of a fixed concentration of masitinib (0, 20, 50 and 100 μM). **b** Masitinib was assayed on eight different dCK substrates including the physiological substrates and several pro-drugs (masitinib/protein ratios: 0, 1, 3, 6, 11 and 22). **c** Kinase inhibitors were investigated to evaluate their effect on substrate phosphorylation (2'dC and gemcitabine) by dCK in the presence of UTP (inhibitor/protein ratio: 22). The experiments were performed in triplicates and data are presented as the mean ± s.d

reduced the number of viable HRT18 cells, whereas the same combination had no effect on the proliferation of Messa 10K.

The anti-tumour effect of masitinib/gemcitabine combination treatment was also investigated in CHO cells (constitutive dCK), CHO dCK- cells (deficient dCK), and CHO dCK Rescue cells (partially reconstituted dCK) (Fig. 1d, e). The CHO cell line showed a high level of dCK expression revealing a strong sensitivity to gemcitabine, which was not affected by the combined masitinib treatment. Conversely, the CHO dCK- cell line deficient in dCK expression did not show any anti-proliferative effect, neither by gemcitabine nor masitinib combined treatment. However, the stable clone selected for low dCK expression

(CHO dCK Rescue) restored the capacity of CHO cells to respond to gemcitabine treatment and produces further anti-tumour benefits when combined with masitinib. These results suggest that the addition of masitinib to gemcitabine treatment induces a decrease in the cell proliferation of the dCK-reconstituted cells, supporting the evidence that masitinib sensitises cancer cells lines to gemcitabine by directly modulating dCK.

**Masitinib modulates nucleoside phosphorylation by dCK**. We also evaluated whether masitinib was able to modulate in vitro gemcitabine phosphorylation directly, using recombinant human

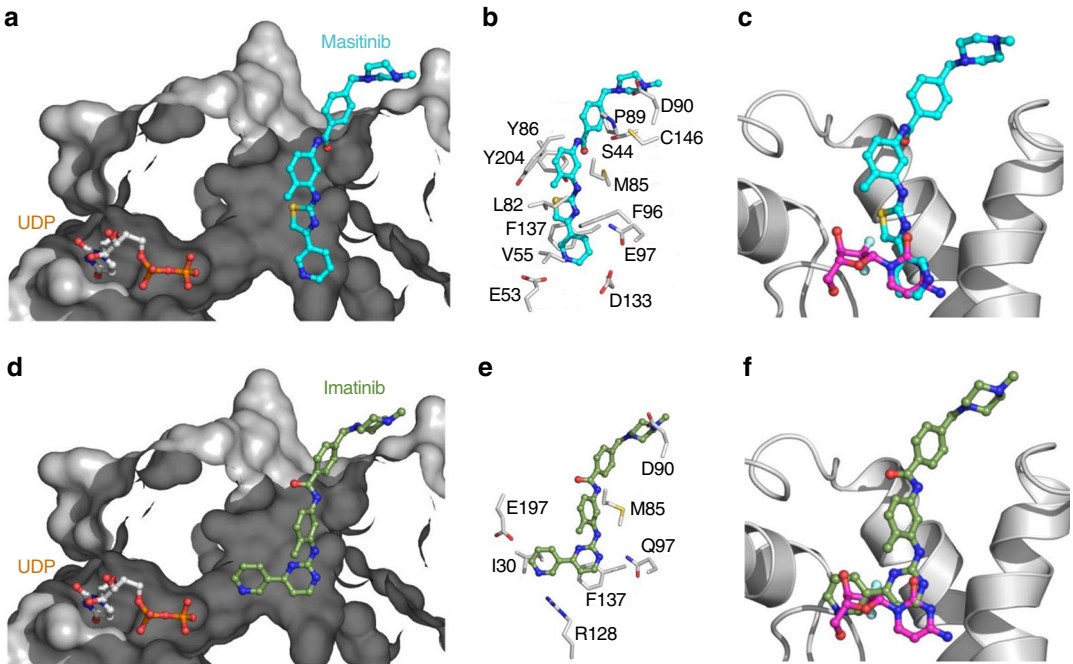

**Fig. 3** Crystal structures of dCK in complex with masitinib or imatinib. Section of the surface representation of the crystal structure of dCK in complex with UDP/masitinib (PDB ID: 5MQL) **a** and with UDP/imatinib (PDB ID: 5MQT) **d**. Detail of the interactions of masitinib (cyan) **b** and imatinib (green) **e** at the active site. Interactions with dCK mainly involve non-polar contacts. Overlay of masitinib **c** and imatinib **f** with gemcitabine (magenta) (PDB ID: 1P62) in the substrate binding site (cartoon representation)

dCK. Optimal conditions for dCK enzymatic assay were determined performing steady-state kinetics experiments in the presence of uridine-5′-triphosphate (UTP), the physiologic phosphate donor of dCK[30, 31]. Increasing concentrations of masitinib led to an augmentation of gemcitabine phosphorylation by dCK (Fig. 2a) increasing $V_{max}$ without affecting the $K_m$ value for UTP (Supplementary Table 2). This effect indicates that masitinib is not a competitor against the phosphate donor, demonstrating a different mechanism of action from c-Kit inhibition[11].

To discern whether the effect of masitinib was selective for gemcitabine, masitinib activation was further challenged using eight different dCK substrates, including the physiological substrates: 2′dC, 2′dA and 2′dG, and several pro-drugs of therapeutic interest: cladribine, fludarabine, cytarabine and decitabine. For each dCK substrate and each concentration of masitinib, the velocity of the reaction was standardised with respect to the drug-free control, and velocity ratios were compared. Figure 2b shows how masitinib activates dCK phosphorylation in a dose-dependent manner in all tested substrates. Activation was more pronounced for deoxy-cytidine-like substrates (pyrimidines), such as gemcitabine and cytarabine. We also tested the putative modulation of masitinib on other cytosolic nucleoside kinases, such as thymidine kinase (TK) and uridine-cytidine kinase 1 (UCK1), but did not observe an enzymatic activation, indicating that masitinib is selective for dCK activation (Supplementary Fig. 6).

**Several TKIs modulate nucleoside phosphorylation by dCK**. Other protein kinase inhibitors, with a structurally different scaffold from masitinib, were then investigated to evaluate their effect on nucleoside phosphorylation by dCK in vitro in the presence of UTP (Fig. 2c, Supplementary Fig. 7). Our data showed that structurally divergent kinase inhibitors were equally capable of modulating dCK activities, albeit for a more limited range of dCK substrates. Among the kinase inhibitors tested, the most active compounds were masitinib, imatinib, tozasertib,

bosutinib and bafetinib. However, the observed effect is not a general class/agent effect, since the majority of kinase inhibitors tested presented relatively low or no activity.

Several clinical studies have evaluated the combined effect of gemcitabine with protein kinase inhibitors, such as axitinib[32, 33] and erlotinib[34, 35]. The latter is approved for the treatment of pancreatic cancer, in association with gemcitabine. It has also been reported that staurosporine may potentiate gemcitabine in vitro on non-small-cells lung cancer cells[36]. Based on such studies, we investigated the effect of axitinib, erlotinib, and staurosporine on dCK enzymatic phosphorylation of gemcitabine (Supplementary Fig. 8). Unlike masitinib, none of the compounds showed an activation effect on dCK enzymatic activity. Consequently, we can assume that the previously observed beneficial effects with axitinib and erlotinib in patients originate from a different mode of action than direct dCK activation.

**Crystal structures of dCK in complex with TKIs**. In order to understand this activation mechanism, we solved crystal structures of dCK in complex with masitinib/UDP and imatinib/UDP (Fig. 3, Supplementary Fig. 9, Supplementary Table 3). We found that the masitinib pyridine ring mimics a pyrimidine base in the nucleoside binding site, while the rest of the molecule extends to a cavity only available in the 'open' conformation of dCK. Previous work has revealed that dCK can adopt several distinct conformations, which are a function of the nature of the nucleoside and nucleotide bound[37]. The designation 'open' or 'closed' relates to the compactness of the nucleoside binding site. The hypothesis for the functionality of the 'open' conformation is that such conformation assists in the initial nucleoside binding and the release of the monophosphate product[37]. The binding mode of imatinib was similar to the one observed for masitinib; however, the pyridine ring in this case was slightly displaced from the nucleoside binding site, as the angle of the pyrimidine moiety (meta) prevents the pyridine ring reaching the bottom of the cavity and the pyridine ring extends to the sugar moiety cavity

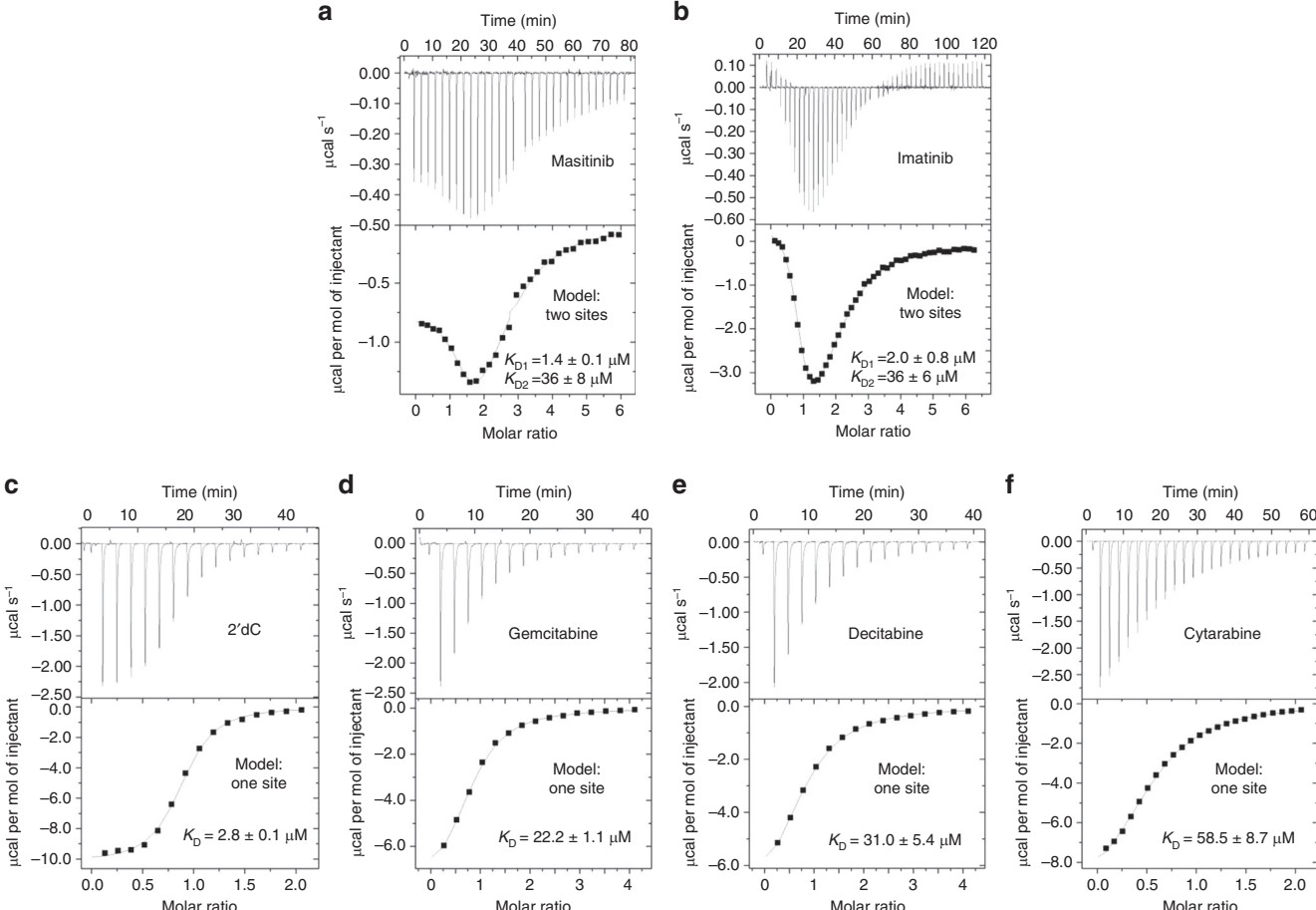

**Fig. 4** dCK binding to protein kinase inhibitors and substrates determined by ITC. Representative ITC titrations for dCK with two protein kinase inhibitors (masitinib **a**, imatinib **b**) and four pyrimidine substrates (2'dC **c**, gemcitabine **d**, decitabine **e**, cytarabine **f**) at 25 °C as a function of the molar ratio of ligand to protein. The upper panel shows the ITC raw data and the lower panel presents the integrated heat of each injection. Binding isotherms were fitted to the raw data using one-site or two-sites binding models as indicated to determine the $K_D$. For substrates, there was one molecule par protein, but for masitinib and imatinib, two binding sites were obtained. Each experiment was performed three times and data are presented as the mean ± s.d

(Fig. 3c, f, Supplementary Fig. 10). This structural observation could explain the less pronounced potentiation effect on dCK enzymatic activity of imatinib *vs.* masitinib (Fig. 2c). For both molecules, interactions with dCK mainly involve non-polar interactions, without direct polar interactions. Additionally, for each ligand-bound structure, a supplementary electron density was present at the top of the cavity (Supplementary Fig. 11), likely corresponding to a second molecule of compound interacting weakly with the first one in a similar binding mode as the 'F-series' inhibitors reported previously[38]. Moreover, it should be noted that masitinib and imatinib were able to open dCK in situ, since we were able to obtain these 3D structures in the 'open' conformation by soaking the protein kinase inhibitors in a 'closed' dCK crystal. The binding of these compounds also released the monophosphorylated substrate (2'-deoxycytidine-5'-monophosphate), observed in the 'closed' dCK crystal and still present in the other nucleoside binding site subunit, elucidating at the same time its mechanism of action.

**Binding affinities of dCK to substrates and TKIs**. To better understand the mechanism of dCK activation, we performed biophysical studies to determine its binding affinities to protein kinase inhibitors and nucleosides by isothermal titration calorimetry (ITC) and bio-layer interferometry (BLI). Masitinib and imatinib direct interaction to dCK was validated by ITC experiments. The dissociation constants of the two binding sites were

determined ($K_{D1} = 1.4\,\mu M$ and $K_{D2} = 36\,\mu M$ for masitinib (Fig. 4a), and $K_{D1} = 2\,\mu M$ and $K_{D2} = 36\,\mu M$ for imatinib (Fig. 4b)), confirming the presence of two compounds per chain, as observed in the crystal structure. In addition, the dissociation constants of four pyrimidine nucleoside substrates (2'dC (Fig. 4c), gemcitabine (Fig. 4d), decitabine (Fig. 4e), and cytarabine (Fig. 4f)) with dCK were determined. Nucleoside studies showed a lower affinity compared to masitinib and imatinib, and a different stoichiometry (1:1), confirming only one binding site. As expected, these protein kinase inhibitors are capable to open dCK due to their higher affinity and are able to displace the monophosphorylated substrate. As the product release is the limiting step for the dCK reaction[37], we assume that these activators are able to better enhance the phosphorylation rate for low affinity substrates, due to the lower affinity of the monophosphorylated substrate. Consequently, activators are able to better compete with and displace the low affinity monophosphate products. As an example, the phosphorylation of the physiological substrate 2'dC is increased by masitinib to a lesser extent (1.9-fold activation, $K_D = 2.8\,\mu M$) than the prodrug gemcitabine (2.9-fold activation, $K_D = 22.2\,\mu M$), due to its higher affinity to dCK.

We further validated these results using BLI technology as an orthogonal assay (Supplementary Fig. 12). The binding of dCK to immobilised biotinylated-masitinib probe was performed (Supplementary Fig. 12c) and competitions assays were used to determine and compare substrates affinity (Supplementary

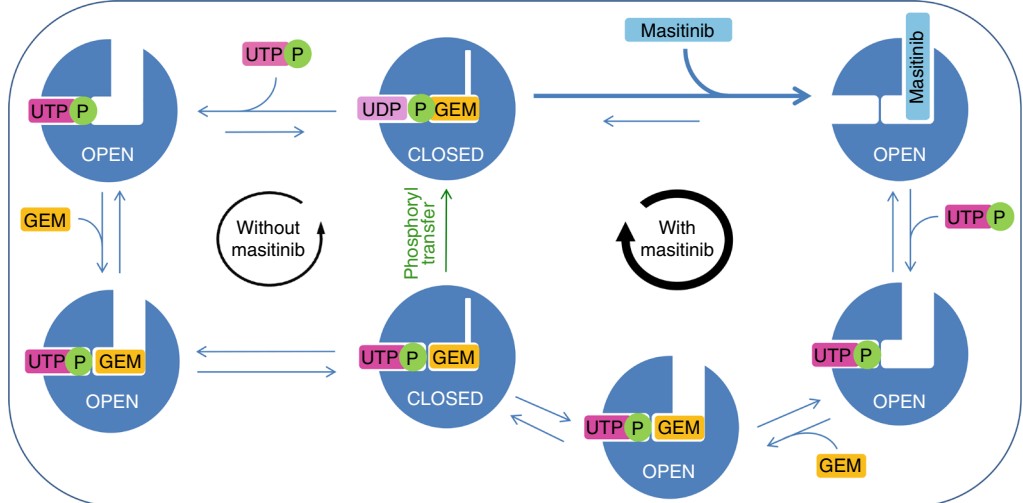

**Fig. 5** Proposed model for masitinib activation. The binding of masitinib on a monophospho-nucleoside-bound enzyme conformation promotes the release of the enzymatic product of the reaction and the transition from the closed state to the open state, being the rate-limiting step for the reaction. After the binding of a fresh molecule of substrate, the enzyme returns to the closed conformation and catalyses the phosphorylation of the nucleoside

Fig. 12d). The dissociation constant determined for dCK/masitinib interaction and substrates affinity confirmed the binding profiles obtained by ITC. We also performed kinetics evaluations of TKIs binding to dCK by performing additional BLI experiment where biotinylated dCK was immobilised (Supplementary Fig. 13). For both compounds, masitinib and imatinib, binding was characterised by a very fast off rate ($k_{off} \sim 10^{-1}\,s^{-1}$) with low residence time ($t_R = 5.7$ and $2.6\,s$ for, respectively, masitinib and imatinib) reflecting the transient nature of the dCK/TKIs complexes (Supplementary Table 4). In order to strengthen our model hypothesis, we sought to compare this kinetics behaviour with a known dCK inhibitor that binds in the same pocket. We synthesised and measured the binding kinetics of a known dCK inhibitor (DI-39)[39, 40] using the same BLI experiment set-up. Although the signal observed for this compound was close to the technical detection limit of the method, we were able to reproducibly measure the binding to dCK and determine the kinetics parameters of this interaction. We measured a slower off rate ($k_{off} \sim 10^{-2}\,s^{-1}$), with a higher residence time ($t_R = 23\,s$), showing a significant difference between activators (masitinib, imatinib) and an inhibitor (DI-39).

## Discussion

An integrative structural and chemical biology program identified dCK as the target protein responsible for the masitinib sensitisation effect on gemcitabine-refractory cancer cells. Further investigation validated an activation mechanism of dCK phosphorylation activity on nucleoside and nucleoside-like pro-drugs in the presence of masitinib. Our findings demonstrate that masitinib, and other TKIs such as imatinib, enhance the dCK-dependent phosphorylation of the pro-drug gemcitabine. In addition, they activate the dCK-dependent phosphorylation of various substrates including the physiological substrates (2′dC, 2′dG, 2′dA) and several pro-drugs of therapeutic interest (e.g., cytarabine, cladribine). Masitinib and imatinib can therefore potentiate the activity of nucleoside analogue agents. Furthermore, our structural data revealed that binding of masitinib or imatinib promotes and maintains an open conformation of dCK, critical for the release of products and the subsequent nucleoside binding[37]. Moreover, we were able to measure the binding kinetics for TKIs to dCK. For both, masitinib and imatinib, binding was characterised by a very fast off rate, reflecting the transient nature of the complex with dCK, with smaller residence times than measured for a known inhibitor. Based on this knowledge we propose a model (Fig. 5) whereby the binding of these TKIs promotes the open conformation and the release of the enzymatic product of the reaction, being the rate-limiting step for the phosphor-transfer. In the absence of substrates, dCK is in equilibrium between an 'open' (dCK-O) conformation and a favoured 'closed' (dCK-C) state[37]. UTP can bind to either the 'open' or 'closed' states, but its binding favours a transition from dCK-C to dCK-O state that is compatible with nucleoside binding. Activators also bind the phospho-nucleoside-bound form of dCK-C and promote the transition from the 'closed' to the 'open' conformation, critical for the release of enzymatic products. This conformational change accelerates the rate-limiting step of the reaction independently of UTP, which permits dCK to bind to the phosphate donor and to continue its enzymatic cycle. After the binding of another molecule of substrate, the enzyme returns to its 'closed' conformation and catalyses a new phosphorylation of the nucleoside. This allows the activator, which presents a better binding affinity to the enzyme than the phospho-nucleoside, to return to the enzymatic pocket, opening the enzyme which then enters into a new cycle.

There are very few examples of mechanistically well characterised small-molecule enzyme activators. These activators include regulators of proteases, kinases, deacetylases, dehydrogenases, phosphatases and nucleases[41]. Four main types of mechanism are listed: the binding of a small molecule to an allosteric site directly in the catalytic domain to promote an active conformation (type A1) (PDK1[42], GK[43]); the binding to an allosteric site to facilitate an irreversible-activating post-translational modification (type A2) (Procaspase-3)[44]; the binding to a regulatory subunit to indirectly promote activity at the catalytic domain (type B1) (AMPK)[45]; and binding to a regulatory subunit to promote an activating oligomerization (type B2) (RNAseL)[46]. Recent examples that illustrate other mechanisms could also lead to small-molecule activation, but they are described as unpredictable and difficult to understand. For aldehyde dehydrogenase 2 (ALDH2), the small molecule activator partially blocks the substrate entrance tunnel, but the substrate access and product release can still occur[47]. Instead of interfering with the enzymatic activity, it leaves the active site residues free to function and increases the enzymatic rate. The authors propose that by binding and blocking one of the active site exits, the activator increases the likelihood of a productive encounter between partners, improving catalytic efficiency. Among the few other examples, we

can cite sirtuin-1 (SIRT1), for which activation with small compounds has been observed but its mechanism remains unclear and is the subject of intense controversy because two opposing models were proposed[48].

dCK activation could be a particular scenario of type A1 activation, as the binding of TKIs in the catalytic domain promote a specific conformation, which increases enzymatic activity. However, the binding site is not purely allosteric, as the bottom of the cavity partially overlaps with the substrate binding. The location of TKIs binding within the substrate binding site of dCK is similar to the binding of a few other molecules such as the 'F-series' (F1, F2, F3, F4)[38] and the DI-39 compound[40], known as potent inhibitors of dCK. The opposite effect of TKIs (activators) and these compounds (inhibitors) on dCK can be partially explained when analysing their crystal structures contact with dCK. Indeed, in the F-inhibitor (or DI-39) bound structure, the catalytic carboxylic acid of the residue Glu53 (catalytic residue responsible for activating the 5′-hydroxyl group of the nucleoside for nucleophilic attack on UTP), makes a 3.2 (or 2) Å hydrogen bond interaction with one of the two exocyclic amine groups[38,40]. These compounds interact directly with an essential active site residue, inhibiting the enzyme by restricting substrate binding and catalysis. In contrast, our structures show that TKIs binds partially at the active site, but without strong direct hydrogen bond interactions and without sterically interfering with the catalytic residues (Fig. 3). Because TKIs only partially block the substrate site with smaller residence time than inhibitors, we suggest that TKI do not interfere with substrates binding and promote an activation of the enzyme activity rather than an inhibition as observed for stronger binders. The very fast off rate observed for TKIs supports our proposed model, where TKIs are capable to bind dCK and promote the release of the reaction product due to their high affinity but they are not competitive inhibitors because of their low residence time and the lack of strong polar interaction with dCK, as shown in the crystal structures. These activators bind the phospho-nucleoside-bound form of dCK-C and promote the transition from the 'closed' to the 'open' conformation, critical for the release of enzymatic products. This conformational change accelerates the rate-limiting step of the reaction independently of UTP, which permits dCK to bind to the phosphate donor and to continue its enzymatic cycle. Although, many more experiments would be necessary to fully validate this model, we truly believe that our conclusions based on this approach combining enzymatic, biophysical and structural data are a step forward in understanding TKIs regulation on dCK activity.

Nucleoside-like drugs are among the most important therapeutic agents currently used to treat tumours and viral diseases. Hence, this property of dCK regulation could be of great therapeutic benefit, by amplifying the effectiveness of dCK-associated therapeutic agents, reducing the toxicity of such agents by maintaining their therapeutic efficiency at lower doses, and/or counteracting the effects of drug resistance initiated via the down modulation of the dCK target. If these observations are confirmed, the combination therapy of small molecule activators plus nucleoside-like anticancer or antiviral agents would represent an innovative therapeutic approach for a variety of diseases.

## Methods

**Reagents and antibodies**. Masitinib (99% pure) and masitinib-NH$_2$ (99% pure) were provided by AB Science S.A. Masitinib-NH$_2$-LC-Biotin was synthesised from masitinib-NH$_2$ (for synthesis and characterisation, see Supplementary Note 1, Supplementary Fig. 14). DI-39 was synthesised as described previously[39, 40]. Protein kinase inhibitors were purchased from Selleckchem. Reagents and substrates for kinetic assays were all purchased from Sigma-Aldrich. Each reagent was obtained as powder and dissolved in H$_2$O or DMSO and stored as aliquots at −20 °C. Fresh dilutions were prepared for each experiment. Primary antibodies used

were a rabbit polyclonal anti-deoxycytidine kinase antibody (ab96599, Abcam, 1:1000), a rabbit polyclonal anti-ERK2 antibody (sc-154, Santa Cruz Biotechnology, 1:2000), a rabbit polyclonal anti-AKT1 antibody (#9272, Cell Signalling Technology, 1:1000), a mouse monoclonal anti-Lyn antibody (610004, BD Biosciences, 1:1000), and a rabbit polyclonal anti-Kit antibody (#3074, Cell Signalling Technology, 1:1000). Primary antibodies were detected using 1:20,000 horseradish peroxidase-conjugated anti-rabbit antibody (Jackson Immunoresearch Laboratories Inc.) or 1:20,000 horseradish peroxidase-conjugated anti-mouse antibody (Dako).

**Cell culture**. The human mast cell leukaemia cell line (HMC-1.1, obtained from M. Arock laboratory, Cachan, France), human ovarian sarcoma cell line (Messa 10 K, obtained from C. Dumontet laboratory, Lyon, France)[27], human colorectal adenocarcinoma cell line (HRT18, obtained from J. Iovanna laboratory, Marseille, France), and human prostatic adenocarcinoma cell line (LNCap, ACC 256, DSMZ) were grown in RPMI 1640 (Gibco). Chinese hamster ovary cells (CHO, CCL-61, ATCC) and dCK deficient cells (CHO dCK-, obtained from M. Meuth, Clinical Research Institute of Montreal, Canada; currently at University of Sheffield, UK) were cultured in DMEM (Dulbecco's modified Eagle medium) (Gibco). CHO cell strains deficient in deoxycytidine kinase activity were selected by isolating dCK-mutants resistant to high concentrations of the nucleoside analogue arabinosyl cytosine (cytarabine). This cell line has no dCK residual levels as confirmed by several techniques at enzymatic and cellular level[28]. Human lung adenocarcinoma epithelial cell line (non-small cell lung cancer) (A549, CCL-185, ATCC) were grown in DMEM/Ham's F12. Cells were tested for mycoplasma contamination using MycoAlert Mycoplasma Detection Kit (Lonza). All cell culture media were supplemented with 10% fetal calf serum, 2 mM ʟ-glutamine, 1% penicillin (10,000 units ml$^{-1}$) and 1% streptomycin (10,000 µg ml$^{-1}$). The cells were maintained in a humidified incubator at 37 °C with 5% CO$_2$.

**Masitinib pull-down**. Masitinib-NH$_2$ was cross-linked via its amino functional group to NHS-activated Sepharose 4 Fast Flow beads according to the manufacturer's instructions (GE Healthcare). Pull-down experiments were performed using HMC-1.1 and HRT18 cell lysates. Briefly, cells were grown until 90% confluence and lysed in 50 mM HEPES, pH 7.5, 1 mM EGTA, 150 mM NaCl, 1.5 mM MgCl$_2$, 10% glycerol, 1% Triton X-100, 0.2% NaF, 1 mM orthovanadate, containing phosphates and proteases inhibitors. Beads cross-linked to the drug (20 µl) or control beads prepared in same conditions without the drug were added to soluble cell lysates (1 ml, 5 mg ml$^{-1}$) and incubated for 2 h at 4 °C. Beads were washed in lysis buffer (50 mM HEPES, pH 7.5, 1 mM EGTA, 150 mM NaCl, 1.5 mM MgCl$_2$, 10% glycerol, 1% Triton X-100). For quality control, 10% of proteins bound to beads were loaded, separated on NuPAGE 4–12% Bis-Tris acrylamide gels in Mops buffer according to the manufacturer's instructions (Invitrogen) and pull-down proteins were visualised after silver staining. For mass spectrometry analysis, pull-down proteins extracts (90%) were loaded on NuPAGE 4–12% Bis-Tris acrylamide gels. Proteins containing bands were stained and visualised with Thermo Scientific Imperial Blue, cut from the gel, and following DTT reduction and iodoacetamide alkylation, digested with high sequencing grade trypsin (Promega). Extracted peptides were concentrated before mass spectrometry analysis.

**Mass spectrometry**. Mass spectrometry analysis was carried out by LC–MS–MS using a LTQ-Velos-Orbitrap (Thermo Electron) connected to a nanoLC Ultimate 3000 Rapid Separation Liquid chromatography system (Dionex). A volume of 5 µl corresponding to 20% of whole sample were injected on the system. After pre-concentration and washing of the sample on a Dionex Acclaim PepMap 100 column (C18, 2 cm × 100 µm i.d. 100 Å pore size, 5 µm particle size), peptides were separated on a Dionex Acclaim PepMap RSLC column (C18, 15 cm × 75 µm i.d., 100 Å, 2 µm particle size) at a flow rate of 300 nl min$^{-1}$ with a two steps linear gradient (4–20% acetonitrile/H$_2$O; 0.1% formic acid for 90 min and 20–45% acetonitrile/H$_2$O; 0.1% formic acid for 30 min). The separation of the peptides was monitored by a UV detector (absorption at 214 nm). For peptides ionisation in the nanospray source, spray voltage was set at 1.4 kV and the capillary temperature at 275 °C. All samples were measured in a data-dependent acquisition mode. The peptide masses were measured in a survey full scan (scan range 300–1700 m/z, with 30 K FWHM resolution at m/z = 400, target AGC value of 1 × 10$^6$ and maximum injection time of 500 ms). In parallel to the high-resolution full scan in the Orbitrap, the data-dependent CID scans of the 10 most intense precursor ions were fragmented and measured in the linear ion trap (normalised collision energy of 35%, activation time of 10 ms, target AGC value of 10$^4$, maximum injection time 100 ms, isolation window 2 Da). Parent masses obtained in orbitrap analyser were automatically calibrated on the 445.120025 ion used as lock mass. The fragment ion masses were measured in the linear ion trap to have a maximum sensitivity and the maximum amount of MS/MS data. Dynamic exclusion was implemented with a repeat count of one and exclusion duration of 30 s. Each sample was analysed in triplicate on the mass spectrometer. Each run was preceded by a blank MS run in order to monitor system background.

**Proteomic analysis**. Raw files generated from mass spectrometry analysis were processed with Proteome Discoverer 1.3 (Thermo fisher Scientific). This software was used to search data via in-house Mascot server (version 2.4.1; Matrix Science Inc.)

against the Human subset (20,264 sequences) of the SwissProt database (version 2014-03). For the database search the following settings were used: a maximum of one miss-cleavage, oxidation as a variable modification of methionine, carbamido-methylation as a fixed modification of cysteine and trypsin was set as the enzyme. A peptide mass tolerance of 6 p.p.m. and a fragment mass tolerance of 0.8 Da were allowed. Only peptides with high stringency Mascot Score were used for protein identification. Peptides FDR <1% was used. Each pull-down experiment was performed using two types of beads to select specific drug interactors from experimental contaminants: beads coupled to the drug and control beads without the drug. A complete list of identified proteins was enclosed in Supplementary Datas 1–4.

To go further in the analysis, relative intensity-based label-free quantification was processed using Progenesis LC–MS software (Progenesis QI for Proteomics v1.0; Nonlinear Dynamics) according to manufacturer instruction (Supplementary Datas 2, 4). First, raw LC Orbitrap MS data were imported and LC–MS heatmap of retention time and m/z were generated. Features from these LC–MS were automatically aligned and filtered to retain signals crossing the following parameters; retention time 15–135 min, m/z 300–1700, charge state 2–6 and 3 or more isotopes. MS–MS spectra were exported into peak list as Mascot generic files (MGF) to allow protein identification using the following inclusion options; the 3 highest intensity precursors for each features and precursor intensity over 25%. MGF files were used to search data via in-house Mascot server (version 2.4.1; Matrix Science Inc.) against the Human database subset of the SwissProt database (version 2014.03). Database search were done using the following settings: a maximum of one trypsin miss-cleavage allowed, methionine oxidation and cysteine carbamido-methylation as fixed modification. A peptide mass tolerance of 6 p.p.m. and a fragment mass tolerance of 0.8 Da were allowed for search analysis. Only peptides adjusted to 5% false discovery rate and with ion score cutoff of 20 were exported from Mascot results and imported back to Progenesis LC–MS for protein grouping and quantification. The raw abundances of all identified peptides were used to normalise LC–MS–MS intensities. Total ion intensity signal from each of the individual peptides generated protein quantification. Any conflicting peptide identifications were removed from the measurements of the quantified proteins. Univariate one-way analyses of variance (ANOVA) were performed within Progenesis LC–MS to calculate the protein p-value according to the sum of the normalised abundances across all runs.

**Cell proliferation assay.** Sensitivity to gemcitabine and combination treatment with gemcitabine and masitinib were investigated by proliferation assays in cell lines using the CellTiter-Blue Reagent (Promega). The cells were plated in 96-well plates (5000 cells per well) in supplemented RPMI-1640 and/or DMEM media and exposed to either an increasing concentration of gemcitabine (0–100 μM) or masitinib (0–10 μM) or the combination of both and incubated for 5 days. At the end of incubation, 10 μl of CellTiter-Blue Reagent were added to each well. The absorbance was read at 590 nm. The effect of the drugs on cells was expressed as a percentage of proliferation compared to untreated cells. The experiments were performed in triplicates and data are presented as the mean ± s.d. Drug synergy effects were analysed using Combenefit (Cancer Research Cambridge Institute), a software tool that enables the visualisation, analysis and quantification of drug combination effects[49]. The data from combination treatments were processed using three synergy reference models: highest single agent (HSA) model, Loewe additivity model and Bliss independence model for HRT18 and LnCAP cell lines.

**Western blot analysis.** The cells were washed in phosphate-buffered saline (PBS) and lysed in HNTG buffer (50 mM HEPES, pH 7, 150 mM NaCl, 1% Triton X-100, 10% glycerol, 1.5 mM MgCl₂, and 1 mM EGTA) containing protease inhibitor mixture (Roche Applied Science), 50 mM NaF, and 100 μM Na₃VO₄. The protein concentration was measured using the Bio-Rad Protein Assay Kit (Bio-Rad Laboratories). The proteins were separated by SDS-PAGE and transferred to polyvinylidene fluoride membrane (Immobilon-P) (Millipore). Membranes were incubated 1 h at room temperature with antibodies and treated using Supersignal West Pico Chemiluminescent Substrate (Pierce). Uncropped blots are shown in Supplementary Fig. 16.

**Construction of pMSCV-dCK vector.** Full-length dCK cDNA was amplified by PCR using dCK-specific primers containing attB sites at their 5′ ends, which allows the rapid and efficient cloning of PCR products into the attP-containing donor vector (pDON R201) using the Gateway technology (Invitrogen). Each PCR product containing vector was then sequenced to verify the dCK sequence integrity. LR recombination reaction was performed between the donor vector and the pMSCV-GW attR-containing destination vector (Clontech Laboratories Inc.).

**Generation of stable CHO clones.** CHO dCK deficient cells (CHO dCK-) were plated in 96-well plates at a density of 5000 cells per well and incubated overnight. The cells were transfected with the pMSCV insert free (vector control) and pMSCV-dCK respectively, using FuGENE 6 Transfection Reagent (Promega) according to manufacturer recommendations. Two days after transfection, cells were plated in 6-well plates and selected in 1 mg ml⁻¹ Geniticin-G418 (Gibco) for 2 weeks. The positive clones were picked and expanded to establish cell lines. The

stable transfection cell clones, named as CHO dCK Rescue, were controlled by western blot analysis for dCK expression levels.

**Cloning and site directed mutagenesis.** dCK cDNA was Gateway cloned into the pDEST17 vector (Invitrogen) from the IMAGE cDNA clone BC103764, leading to the expression of the NH₂-hexahistidine-tagged full length enzyme. The pDEST17 expression vector encoding for dCK was used for site directed mutagenesis in order to increase protein stability by mutating three of the six exposed cysteines (C9S, C45S and C59S; named as dCK-C3S), as previously described[50]. The pDEST17 expression vector encoding for dCK-C3S-S74E was obtained in the same way as dCK-C3S (the S74E mutation mimicking a phosphorylated state)[37]. All mutations were confirmed by DNA sequencing. Primers used for the different mutations are listed in Supplementary Table 5. UCK1 isoform 1 (XM00005272224.1) cDNA was obtained by gene synthesis (Geneart, subcloned into pDONR 221) and cloned into pET 61-DEST (Novagen) by Gateway LR recombination. TK1 (BC006484) cDNA was cloned into pDEST17 AA 2-end by Gateway LR recombination.

**Protein expression and purification for kinetic assays.** BL21 AI Escherichia coli cells transformed with dCK plasmid were grown in LB media containing 100 μg ml⁻¹ ampicillin at 37 °C for 4 h after arabinose induction. BL21 DE3 E. coli cells transformed with UCK1 plasmid were grown in 2XYT media containing 100 μg ml⁻¹ ampicillin at 12 °C for 20 h after induction with 1 mM IPTG. The cells transformed with TK1 plasmid were grown in 2XYT media containing 100 μg ml⁻¹ ampicillin at 37 °C for 4 h after arabinose induction.

The cells were re-suspended in 20 mM TRIS, pH 8, 500 mM NaCl, 0.1% BRIJ-35 and 5% glycerol buffer and EDTA free anti-protease was added following manufacturer recommendation (Thermo Fisher Scientific). After sonication and centrifugation (30,000×g), the supernatant was loaded on a 5 mL HisTrap FF column (GE healthcare) pre-equilibrated with 20 mM TRIS pH 8, 500 mM NaCl, 0.01% BRIJ-35 and 5% glycerol buffer. The protein was then eluted by the same buffer containing 250 mM Imidazole. The eluted fraction was applied on a HiPrep 26/10 desalting column (GE healthcare) pre-equilibrated with 20 mM TRIS pH 8, 500 mM NaCl, 5% glycerol and 0.01% BRIJ-35 buffer. Finally, the protein at 20% glycerol was flash frozen in liquid nitrogen and stored at −80 °C.

**dCK and UCK1 kinetic assay.** Kinetic analyses were performed using a spectrophotometric continuous enzymatic-coupled assay using pyruvate kinase (PK) and lactate dehydrogenase (LDH). Analysis of the effect of masitinib on dCK (WT, C3S and C3S-S74E) activity was performed using UTP as phosphate donor, as UTP is the physiological dCK-phosphate donor[30]. dCK mutants presented the same enhanced nucleoside phosphorylation produced by masitinib as the wild-type (Supplementary Fig. 15). Analysis of the effect of masitinib on UCK1 activity was performed using ATP as phosphate donor. PK/LDH coupled assay is based on the conversion of phosphoenolpyruvate (PEP) and UDP/ADP to pyruvate and UTP/ATP by PK and the subsequent conversion of pyruvate to lactate by LDH. The latter step requires NADH, which is oxidised to NAD⁺. NADH is fluorescent (excitation at 337 nm and emission at 460 nm), but not NAD⁺. Thus, the measurement of fluorescent decrease at 460 nm is a measure of kinase activity. dCK experiments were performed in 50 mM HEPES, pH 7.5, 5 mM MgCl₂, 1 mM DTT, and 0.01% BRIJ-35 buffer supplemented by dCK (9 μM), substrate (1 mM), UTP (2 mM), and masitinib at varying concentrations. UCK1 experiments were performed in 50 mM TRIS, pH 7.6, 100 mM KCl, and 5 mM MgCl₂ buffer supplemented by UCK1 (1 μM), substrate (2 mM), ATP (500 μM), and masitinib at varying concentrations. All measurements were performed on a BMG Labtech Pherastar FS apparatus. All assays were performed in triplicate and each experiment was performed at least twice. The data are presented as mean ± s.d. The results were plotted as kinetics and the velocity was determined for each condition as the slope of the linear range of the curve. For each substrate and each concentration of masitinib, the velocity of the reaction (according to $V = d[P]/dt$) was standardised with respect to the drug free control and velocity ratios were compared. $K_m$ (Michaelis–Menten constant) and $V_{max}$ (maximal velocity) values were determined using PRISM software (GraphPad Software Inc.) by fitting the experimental data according to the Michaelis–Menten approximation defined as Equation 1.

$$V = (V_{max} \cdot [S]) / (K_m + [S]) \tag{1}$$

**TK1 kinetic assay.** The analysis of the effect of masitinib on TK1 activity using ATP as phosphate donor was assayed with the HTRF Transcreener ADP assay (Cisbio International), an immuno-assay based on the competition between the native ADP (generated by the reaction of transfer of phosphate catalysed by TK1) and a fluorescent tracer, the ADP-d2. ADP and ADP-d2 compete for the binding to a monoclonal anti-ADP antibody labelled with Europium (Eu³⁺) cryptate. This assay comprises two steps: first, an enzymatic step, during which the substrate is incubated with TK1 in the presence of ATP and Mg²⁺ leading to the generation of native ADP, and second, at the end of the reaction (stopped by addition of EDTA which chelates Mg²⁺), the antibody anti-ADP-Eu³⁺ (emission 620 nm) is added in the presence of the fluorescent tracer ADP-d2 (emission 665 nm). The obtained

signal is inversely proportional to the concentration of ADP in the sample. All measurements were performed on a BMG Labtech Pherastar FS apparatus. Results are expressed in delta fluorescence ($\Delta F$) unit defined as follow $\Delta F\% = [(ratio - ratio\ blank)/(ratio\ blank)] \times 100$, where $ratio = (665\ nm/620\ nm) \times 10^4$.

**Protein expression and purification for structural studies.** C41 (DE3) PLysS *E. coli* cells transformed with *dCK*-plasmids (dCK-C3S (crystallisation assays) and dCK-C3S-S74E (ITC and BLI assays)) were grown in LB media containing 100 µg ml$^{-1}$ ampicillin and 34 µg ml$^{-1}$ chloramphenicol at 37 °C for 4 h after induction with 0.1 mM IPTG. The cells were re-suspended in 50 mM TRIS, pH 8, 500 mM NaCl, 30 mM imidazole and 10% glycerol buffer and EDTA free anti-protease was added (Thermo Fisher Scientific). After sonication and centrifugation (30,000×*g*), the supernatant was loaded on a 5 ml HisTrap FF column (GE healthcare) pre-equilibrated with re-suspension buffer. Proteins were then eluted by re-suspension buffer containing 500 mM imidazole. Eluted fractions were mixed with TEV protease and dialysed overnight at 4 °C against 50 mM TRIS pH 8, 150 mM NaCl and 10 mM DTT buffer. After cleavage, Ni-NTA beads were used to retain the non-cleaved fraction and recover the cleaved dCK. dCK was further purified by size exclusion chromatography (superdex 200 16/600, GE healthcare) in 20 mM HEPES, pH 7.5, and 200 mM NaCl before concentration to 12 mg ml$^{-1}$ and storage at −80 °C.

**Crystallisation and soaking.** dCK-C3S used for crystallisation assays contained three mutations in order to generate better quality crystals (three solvent-exposed cysteine residues were mutated to serines; C9S, C45S and C59S). Structures in complex with nucleotides were determined using the diphosphate form since co-crystallisation with the triphosphate would result in the enzymatic reaction taking place. Before crystallisation assays, UDP was added to the protein stock at 5 mM final concentration. Needle crystals of apo dCK-C3S were obtained using hanging drop vapour diffusion at 19 °C or 12 °C. For the masitinib complex, drops were constituted of 1 µl of protein/UDP mix and 1 µl of the reservoir solution containing 100 mM sodium acetate pH 5.5, 100 mM ammonium sulphate, and 20% PEG 3350 (w/v). For imatinib complex, drops were constituted of 1.5 µl of protein/UDP mix and 1 µl of the reservoir solution containing 100 mM sodium acetate pH 6, 13% PEG 4000 (w/v), and 6% isopropanol. One-week-old crystals were soaked, for 24 h, in a drop constituted of 1 µl of reservoir buffer and 1 µl of 20 mM masitinib dissolved in protein buffer or, for 18 h, in a drop of 2 µl of reservoir buffer containing 10 mM imatinib.

**Data collection and processing.** Prior to data collection, crystals were rapidly soaked in a cryo-buffer, containing reservoir solution complemented with 25% (v/v) glycerol, and subsequently flash frozen in liquid nitrogen. X-ray data were collected at the European Synchrotron Radiation Facility (Grenoble, FRANCE) on beamline ID23-1 and ID23-2. The data reduction and scaling were performed using XDS suite[51].

**Structure determination and refinement.** The structure of dCK-C3S was solved by molecular replacement using PHASER[52] and the chain A of the recently published dCK-C4S structure (PDB ID: 1P60)[53] was used as starting model. The initial molecular replacement solution was further refined using BUSTER (Bricogne G., Blanc E., Brandl M., Flensburg C., Keller P., Paciorek W., Roversi P., Sharff A., Smart O.S., Vonrhein C., Womack T.O. (2011). BUSTER version 2.10.2. Cambridge, United Kingdom: Global Phasing Ltd.), alternating with cycles of manual rebuilding in COOT[54]. Full data and refinement statistics are shown in Supplementary Table 3.

**Isothermal titration calorimetry.** ITC was used to evaluate the binding between dCK-C3S-S74E and the selected compounds. The S74E mutation mimics the phosphorylated state of this serine, which favours the open conformation of the enzyme, making it competent for nucleoside binding[37] to evaluate the binding affinity with the selected compounds. Purified dCK-C3S-S74E and the compounds were diluted in the ITC buffer containing 20 mM HEPES, pH 7.5, and 200 mM NaCl. Titrations were carried out on a MicroCal ITC200 microcalorimeter (GE Healthcare) at 25 °C using 16 or 24 injections of the titrant (1–3 mM compound in the syringe) into the analyte (100 or 300 µM protein in the cell). A first small injection (0.2 µl) was included in the titration protocol in order to remove air bubbles trapped in the syringe prior titration. Because of the low solubility of several ligands, sequential titrations were employed to achieve sufficient molar ratios for reliable model fitting, and these titrations were linked together prior to data analysis using ConCat32 software (Microcal). Raw data were scaled after setting the zero to the titration saturation heat value. Integrated raw ITC data were fitted to a one site (substrates) or two sites (masitinib and imatinib) nonlinear least squares fit model using MicroCal Origin 9.1 (Origin Lab). Each experiment was performed three times and data are presented as the mean ± s.d.

**Bio-layer interferometry.** The binding affinity between dCK-C3S-S74E and masitinib was determined by Bio-Layer Interferometry (BLI). Masitinib-NH$_2$-LC-biotin conjugate containing a glycol spacer moiety (Supplementary Note 1) was immobilised on Super Streptavidin Biosensors (SSA) (ForteBio) at a concentration

of 1 µM in a buffer containing 10 mM sodium acetate pH 6, and 0.5% DMSO. Binding of serially diluted dCK-C3S-S74E (0.1–50 µM) was measured using 5 min association and 5 min dissociation steps at 30 °C with a shake speed of 1000 r.p.m. All measurements were performed in binding buffer containing 20 mM HEPES, pH 7.5, 200 mM NaCl and 0.1% BSA. Sensorgrams were measured on an Octet RED96 system (ForteBio) and referenced against the reference sensors signals using the Octet Data Analysis 8.2 software (ForteBio). The equilibrium dissociation constant ($K_D$) was determined at equilibrium using the steady-state analysis. Competition assays with ligands were achieved using the same methodology. Binding of dCK-C3S-S74E (10 µM) to immobilised masitinib was measured in the presence or absence of competitors (100 µM for substrates and 20 µM for masitinib and imatinib). Percentage of competition was obtained after normalisation with the control without competitor (dCK-C3S-S74E alone, 0% of competition). Each experiment was performed three times and the data are presented as the mean ± s.d.

BLI was also used to measure the kinetics parameters for dCK binding to TKIs and DI-39. dCK-C3S-S74E (2 mg ml$^{-1}$) was biotinylated by incubation with NHS-PEG4-biotin (Thermo Scientific) in a 1:1 molar ratio for 30 min at room temperature and 1.5 h at 14 °C on a rocking platform. Excess unreacted biotinylation reagent was removed by passage through a 5 ml Zeba Spin Desalting Column (Thermo Scientific). BLI measurements were carried out using an Octet RED96 instrument (ForteBio). Biotinylated dCK was immobilised on SSA biosensors at a concentration of 5 µM after 1200 s. Free streptavidin sites were blocked by incubating with 10 µg ml$^{-1}$ biocytin (Sigma-Aldrich) for 100 s and 1% BSA for 1000 s. A control set of SSA biosensors were prepared in parallel by blocking the surface with biocytin instead of protein. A three-point concentration series were prepared for each compound and were measured using 60 s association and 140 s dissociation steps at 30 °C with a shake speed of 1000 r.p.m. All measurements were performed in binding buffer containing 20 mM HEPES, pH 7.5, 200 mM NaCl, 0.1% BSA and 1% DMSO. Each experiment was performed at least twice and $K_D$, $k_{on}$ and $k_{off}$ values were determined using the global fitting procedure in the ForteBio Data Analysis Software. Data are presented as the mean ± s.d. Residence times were calculated from values of $k_{off}$ ($t_R = 1/k_{off}$).

**Data availability.** Structures have been deposited in the Protein Data Bank under accession codes 5MQJ (dCK-C3S), 5MQL (dCK-C3S in complex with masitinib) and 5MQT (dCK-C3S in complex with imatinib). The mass spectrometry proteomics data have been deposited to the ProteomeXchange Consortium with the data set identifier PXD007765. The data that support the findings of this study are available within the paper and its Supplementary Information files or available from the corresponding author upon reasonable request.

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

## Acknowledgements

This work was supported by the Fondation ARC pour la recherche sur le cancer (grant n° SL220130606659). We thank the staff of ESRF and EMBL-Grenoble for assistance and support in using beamlines ID23-1 and ID23-2. We thank Athel Cornish-Bowden and María Luz Cárdenas for helpful discussions and their valuable advice on enzymatic assays. We are thankful to Pascaline Camail who participated as an undergraduate program intern.

## Author contributions

P.D. and X.M. conceived the project. K.H., L.G., S.Lo. and S.Le. performed molecular biology and biochemistry experiments. E.B. and S.A. performed chemoproteomic assays. K.H., M.H., S.Le. and N.C. performed cellular assays. M.S.-A., L.G. and B.H. performed enzymatic assays. M.S.-A., E.R. and A.L. performed crystallography. M.S.-A. and S.B. performed biophysical experiments. S.C. synthesised compounds. All authors were involved in data interpretation. K.H., M.S.-A., E.R., S.B., C.M., A.M., P.D.S., X.M. and P. D. wrote the manuscript.

## Additional information

**Competing interests:** L.G., B.H., M.H., S.Le., N.C., C.M., A.M. and P.D. are shareholders of AB Science. The remaining authors declare no competing financial interests.

