## [Peer Review File · Nature Communications]

Reviewers' comments:

Reviewer #1 (Remarks to the Author):

The authors follow up on an observation that showed a striking synergy of the selective tyrosine kinase inhibitor (TKI) masitinib with gemcitabine on different cancer cell lines. To study the mechanistic basis for these observations, chemical proteomics experiments with masitinib derivatives that could be covalently bound to beads were performed and showed a strong enrichment for deoxycytidine kinase (dCK) that is a key enzyme to activate gemcitabine and other nucleoside analogues. Loss-of-function experiments of dCK showed loss in gemcitabine sensitivity. Detailed enzymatic experiments showed that dCK is activated by masitinib (through an increased v_{max}). The authors convincingly demonstrate that masitinib is able to activate dCK-dependent activation different natural and pro-drug substrates of dCK and that several other TKIs, including imatinib, showed the same effect on dCK activity, whereas other TKIs were not acting as dCK activators. Two crystal structures of dCK in complex with UDP/masitinib and UDP/imatinib showed binding of the two TKIs to the nucleoside binding site with evidence for a secondary binding site. ITC and BLI experiments showed low micromolar binding affinities for both TKIs and mid-micromolar binding to a second site. The authors conclude with a model in which they propose that masitinib activates dCK by facilitating the rate-limiting product release step.

Overall, this is a nice paper that reports on a new mode-of-action of a subgroup of TKIs and a critical mechanism that determines the activation of nucleoside analogue pro-drugs. The manuscript is well written and easy to follow. The experiments are all well controlled and clearly presented. This manuscript will certainly be of interest to a broad readership.

There are some points that I would like to ask the authors to address to improve the manuscript:

1. lines 99-101: The authors should show the data that 'masitinib had no significant anti-proliferative activity' on the three cell lines shown in Fig. 1a.
2. lines 103-118: It would be great to include a brief summary of the MS results: As one would expect that besides the off-target dCK, the on-target(s) for masitinib and other kinase off-targets were identified, could the authors list those? How many kinases were identified at which abundance?

3. Figure 1e: The employed cell lines are not well described: What are the CHO dCK- cells: Are these shRNA or CRISPR knock-down? Please clarify.

4. Figure 1e: CHO cells do not seem to be the best choice of for these experiments, as they do not show the strongly increase gemcitabine sensitivity upon masitinib treatment, but are already exquisitely sensitive in the absence of masitinib - in contrast to the cells in Fig. 1a. Therefore, while the genetic knock-down and rescue experiments are convincing, would it not be expected that masitinib should have an effect on the knock-down cells (with possible residual dCK levels) or the rescued cells (with low dCK levels)? Why were not HRT18, A549 or LnCAP cells used for the dCK knock-down and rescue experiments?

5. Figure 3: It would be insightful to highlight in a figure where the additional electron density for the secondary TKI binding site is located.

6. lines 203-206: A figure should illustrate what is being described here.

7. Finally, I am not too convinced of the model that the authors propose: The binding affinities of masitinib and imatinib to dCK are low micromolar and their binding site is the nucleoside binding pocket. Hence, masitinib and imatinib should rather act as competitive inhibitors than activators. But I guess that based on the data that is available, it will be difficult to really verify the proposed model. For this, kinetic data, including k_{on} and k_{off} rates for both substrates and TKIs would be necessary and the residence times would need to be measured. Without this, the model that masitinib promotes product release remains a pure speculation that is not supported by data. Revise and tone-down claims.

Reviewer #2 (Remarks to the Author):

In the work by Hammam et al an interesting mechanism is discovered potentially explaining synergies between masitinib and gemcitabine. Using affinity proteomics they discover an interaction between deoxycytidine kinase (dCK) and masitinib that is confirmed by biophysical and structural studies. Although masitinib binding overlap with the substrate nucleoside binding site they propose that the ligand serve as an activator of dCK, rather than an inhibitor. Using activity studies of a subset of nucleoside substrates, including drugs, they provide support for that this might constitute a general activation mechanism of dCK. Furthermore, they show that there is a correlation between dCK expression and potentially synergistic effects in cell lines.

Overall the work is quite well presented and the combined data supports the conclusions that this mechanism is contributing to the synergies seen for the two drugs in cell lines. I think this

discovery at the cell line level, which also has implications for the development of dCK activating drugs, is enough to merit the work for publication in Nature Communications, in spite of that other mechanism might be operative for the synergies seen in some clinical studies.

Still, there are quantitative aspects of the effect of this activation mechanism which would be interesting to shed further light on and, for example, measurement of the changes in intracellular levels of the active adduct of gemcitabine would strengthen the work very significantly (e.g. see methods from Per Artursson's group).

I also have problems with one of the quantifications in figure 1. In fig 1c it is shown the activation of gemcitabine (at inhibitor/protein ratio 22) is 3 times stronger than for dC. However, if you compare gemcitabine at an inhibitor/protein ratio of 22 in fig1b it is only 1.5 times stronger. For the mechanism this makes little difference but why is the data not consistent?

Concerning the somewhat controversial activation mechanism where the binding is overlapping with the substrate-binding site, and presumably assisting in releasing the product/opening the active site, the authors should take the discussion a bit further. For example, there are probably other enzymes with activation mechanisms which can serve as a precedent for this type of scenario, and they should find references for those.

Overall, it looks like the manuscript needs a bit more work and there are also a number of statements in the paper referring to literature data but where the reference is missing.

Response to Reviewers' comments:

Reviewer #1 (Remarks to the Author):

The authors follow up on an observation that showed a striking synergy of the selective tyrosine kinase inhibitor (TKI) masitinib with gemcitabine on different cancer cell lines. To study the mechanistic basis for these observations, chemical proteomics experiments with masitinib derivatives that could be covalently bound to beads were performed and showed a strong enrichment for deoxycytidine kinase (dCK) that is a key enzyme to activate gemcitabine and other nucleoside analogues. Loss-of-function experiments of dCK showed loss in gemcitabine sensitivity. Detailed enzymatic experiments showed that dCK is activated by masitinib (through an increased v_{max}). The authors convincingly demonstrate that masitinib is able to activate dCK-dependent activation different natural and pro-drug substrates of dCK and that several other TKIs, including imatinib, showed the same effect on dCK activity, whereas other TKIs were not acting as dCK activators. Two crystal structures of dCK in complex with UDP/masitinib and UDP/imatinib showed binding of the two TKIs to the nucleoside binding site with evidence for a secondary binding site. ITC and BLI experiments showed low micromolar binding affinities for both TKIs and mid-micromolar binding to a second site. The authors conclude with a model in which they propose that masitinib activates dCK by facilitating the rate-limiting product release step.

Overall, this is a nice paper that reports on a new mode-of-action of a subgroup of TKIs and a critical mechanism that determines the activation of nucleoside analogue pro-drugs. The manuscript is well written and easy to follow. The experiments are all well controlled and clearly presented. This manuscript will certainly be of interest to a broad readership.

There are some points that I would like to ask the authors to address to improve the manuscript:

1. lines 99-101: The authors should show the data that 'masitinib had no significant anti-proliferative activity' on the three cell lines shown in Fig. 1a.

Requested data are shown in Supplementary Figure 1 and an analysis to confirm the synergistic effect has been performed using Combeneft (Cancer Research Cambridge Institute), a software tool that enables the visualization, analysis and quantification of drug combination effects¹. Data from combination assays have been processed using three classical Synergy reference models: Highest single agent (HSA) model, Loewe additivity model and Bliss independence model for HRT18 and LnCAP cell lines. The three models confirm a synergistic effect for Masitinib and Gemcitabine (Supplementary Figure 2). The sentence in lines 99-101 has been changed for "In contrast, single-agent masitinib had only a slight anti-proliferative activity in these pathologies (Supplementary Fig. 1), indicating a synergistic interaction for the combined treatment (Supplementary Fig. 2)".

2. lines 103-118: It would be great to include a brief summary of the MS results: As one would expect that besides the off-target dCK, the on-target(s) for masitinib and other kinase off-targets were identified, could the authors list those? How many kinases were identified at which abundance?

In the Supplementary Information, there are four excel files with the identification of masitinib targets by mass spectrometry on HMC-1.1 (Supplementary Data 1) and HRT18 cellular lysates (Supplementary Data 3) and the relative quantification of masitinib associated proteins on HMC-1.1 (Supplementary Data 2) and HRT18 cellular lysates (Supplementary Data 4) used to generate the Volcano plot (Fig 1b). We have highlighted in the quantification files all the kinases with the larger fold changes that are also statistically significant. We have added a paragraph in lines 114-130 outlining a brief summary of the Mass Spectrometry results and, a table showing the on-target and off-target kinases with their abundances (Supplementary Table 1).

"The abundance of proteins relative to the control and statistical significance were calculated (Supplementary Note 2). A complete list of the identified proteins and their quantification is enclosed in Supplementary Data 1-4 and a table showing the on-target and off-target kinases with their abundances is presented in Supplementary Table 1. We identified the masitinib known targets c-Kit and Lyn in HMC-1.1 cellular lysates, a cell line expressing both proteins, and only Lyn in HRT18 cellular lysates, as this cell lines does not express c-Kit (Supplementary Table 1). We only identified three other kinases with low abundance, certainly due to the high selectivity profile of masitinib towards c-Kit²: the mitochondrial acylglycerol kinase (AGK), the catalytic subunit of the DNA-dependent protein kinase (DNA-PKcs) and the Abelson tyrosine-protein kinase 2 (ABL2)."
AGK is a mitochondrial membrane protein with lipid kinase activity which phosphorylates monoacylglycerol and diacylglycerol to form lysophosphatidic acid and phosphatidic acid. DNA-PKcs is a serine/threonine protein kinase that plays an important role in the DNA damage response and maintenance of genomic stability. ABL2 is a tyrosine-protein kinase that plays an ABL1-overlapping role in key processes linked to cell growth and survival.

"This result is not surprising as these kinases use ATP as phosphate donor and masitinib is an ATP mimic. We also identified the NAD(P)H:quinone oxidoreductase NQO2 which was also found as one of the foremost interactors of other TKIs on the same type of reverse proteomics approach³. We rapidly identified deoxycytidine kinase among the list of potential candidates (Fig. 1b). This protein was identified with a high score ratio and

was listed 3rd in the HMC-1.1 cell line cellular lysates in the fold change relative to control value ranking (Fig. 1b). When using the HRT18 cell line, dCK was ranked 27th which supports our previous result as this cell line has a low dCK expression (Fig. 1c). These data prompted us to evaluate dCK as the protein vector responsible for the masitinib sensitisation effect on gemcitabine-refractory cancer cells.”

3. Figure 1e: The employed cell lines are not well described: What are the CHO dCK- cells: Are these shRNA or CRISPR knock-down? Please clarify.

We have properly described the origin of CHO dCK- cells in the Methods section (lines 474-480). The Chinese hamster ovary dCK deficient cells (CHO dCK-) were obtained from M. Meuth laboratory (Clinical Research Institute of Montreal, Canada; currently at University of Sheffield, UK). CHO cell strains deficient in deoxycytidine kinase activity were selected by isolating the dCK- mutants resistant to high concentrations of the nucleoside analogue arabinosyl cytosine (cytarabine) as published previously⁴. This cell line has no dCK residual levels as confirmed by several techniques at enzymatic and cellular level.

4. Figure 1e: CHO cells do not seem to be the best choice of for these experiments, as they do not show the strongly increase gemcitabine sensitivity upon masitinib treatment, but are already exquisitely sensitive in the absence of masitinib - in contrast to the cells in Fig. 1a. Therefore, while the genetic knock-down and rescue experiments are convincing, would it not be expected that masitinib should have an effect on the knock-down cells (with possible residual dCK levels) or the rescued cells (with low dCK levels)? Why were not HRT18, A549 or LnCAP cells used for the dCK knock-down and rescue experiments?

Reviewer 1 proposed to use resistant cell lines (as illustrated Fig. 1a) which could be assessed with combined treatment (Gemcitabine+Masitinib) after dCK knock-down and rescue experiments. We agree with reviewer comment that the result obtained with CHO cell line is not completely satisfactory. However, we will first explain why we chose CHO cell line and we will try then to explain through new experiments that using other cell lines could not yield to a meaningful outcome.

We chose CHO cell lines in regard to their high clonability efficacy and the availability of dCK- mutants.

Indeed, we used sensitive cell lines (CHO cell line) that became resistant after dCK deletion (CHO dCK-deficient) referenced in the literature to show a complete loss of dCK expression^{4,5} (and our western blot presented in supplementary data Figure 5a). We reconstituted dCK expression in deficient CHO cell line by two different methods either by transfection or by lentiviral infection. The problem of such experiments is that after reconstitution, CHO dCK reconstituted was highly sensitive to gemcitabine (as high as the parental cell line) and no synergy was observed in the presence of masitinib. To note, the synergic response is only observed with a “low expression status” of dCK.

Western blotting control of the reconstitution of dCK in CHO dCK- cell line is presented below. HMC1 and parental CHO were used as positive control and CHO dCK- as negative control. ERK2 antibody was used as the loading control.

As illustrated on the figure below, CHO reconstituted with dCK (green) is even more sensitive to gemcitabine than parental CHO cells (blue), whereas CHO dCK – (red) are resistant as expected. To note, no gain of gemcitabine sensitivity has been obtained with this reconstituted cell line in further experiments using masitinib since a very high level of dCK expression is incompatible with synergistic response.

We thus decided to obtain clones from dCK-transfected CHO deficient cells in order to get different levels of dCK expression (characterization of 30 clones). Then, each clone was analyzed for dCK protein expression by western blot. Figure below represents one of this clones selected on the basis of “low” dCK expression (lower band) (G4-24 is a dCK negative clone. G4-6 and G4-25 have low expression of dCK).

After selection of low expression clones, we assessed each clone for gemcitabine sensitivity. On the figure presented above, we can observe one of these clones (clone G4-6) (purple) which has an intermediate response for gemcitabine sensitivity as compared to gemcitabine response with both CHO parental and deficient cell lines. We have shown in the Figure 1e of the manuscript, the effect of masitinib on the G4-6 clone as “CHO dCK Rescue” cells. We observed a comparable effect of masitinib in gemcitabine sensitivity (even if lighter) for the G4-25 clone.

In order to completely answer to reviewer #1 and editor request, we developed another cell line model for which a sensitive and a resistant cell line were immediately available.

We used MESSA cell line (ovarian cancer cells) which is sensitive to gemcitabine and its resistant counterpart MESSA 10K which is dCK deficient⁶. After dCK reconstitution (by lentiviral infection), MESSA 10K +dCK shows again a high recovery for gemcitabine sensitivity (comparable to original sensitivity obtained with MESSA cells). We thus decided to FACS-cell sorting MESSA10K +dCK rescued cells in function to dCK expression level (see figure below) (i.e. GFP co-expression thanks to a bicistronic lentiviral vector dCK IRES GFP). Thus, we characterized four different populations in regard to dCK expression. We named them dCK HIGH / Medium HIGH / Medium LOW and LOW.

Then, we confirmed dCK expression level by western blot on the four cell sorted MESSA 10K + dCK populations as compared to MESSA parental cell line and its resistant counterpart MESSA 10K.

As we can see on the western blot above, MESSA 10K do not express any dCK protein as compared to the parental MESSA cell line (as already described⁶). In our reconstituted cell lines we obtained a noteworthy correlation between GFP expression and dCK expression level. To note, even by trying to cell sort a population with the lowest possible level of GFP/dCK expression (MESSA 10K dCK Low), dCK level was comparable to the parental sensitive MESSA cell line. It was impossible in our hand to characterize a population with a lower dCK level compatible with a strong gemcitabine resistance.

As expected, when assessed against gemcitabine, all four dCK reconstituted populations showed gemcitabine sensitivity even if some slight differences could be observed between the different populations.

In spite of this sensitivity phenotype we challenged each MESSA population for masitinib synergy as you can see on the proliferation assay presented below. We obtained (as for CHO rescue experiments) only a slight effect with masitinib only in the MESSA 10K +dCK LOW population with the lowest doses of Gemcitabine (comparable with what we obtained with the parental MESSA cell line).

In regard of our new data, we feel that all attempts to delete dCK gene in cells followed by a dCK rescue will give rise to the same result, *i.e.* a full reconstitution of the gemcitabine sensitivity incompatible with this subtle synergy phenomenon.

5. Figure 3: It would be insightful to highlight in a figure where the additional electron density for the secondary TKI binding site is located.

A figure showing the additional electron density for the second molecule of masitinib and imatinib has been added in supplementary information (Supplementary Figure 10).

Supplementary Figure 10

6. lines 203-206: A figure should illustrate what is being described here.

We have added in the text the number of the figures that illustrate the result described in lines 216-219. As shown in Figures 3c and 3f, where masitinib and imatinib are superposed with gemcitabine, the masitinib pyridine ring mimics the pyrimidine base of gemcitabine in the nucleoside binding site; however, the pyridine ring in the case of imatinib was slightly displaced from the nucleoside binding site. The angle of the pyrimidine moiety (meta) prevents the pyridine ring to reach the bottom of the cavity and the pyridine ring extends to the sugar moiety cavity. We have added an extra figure showing the superposition of both compounds to illustrate this difference (Supplementary Figure 9). This structural observation, where masitinib is able to extend deeper in the cavity than imatinib, could explain the less pronounced potentiation effect on dCK enzymatic activity of imatinib vs. masitinib for both substrates, as shown in Figure 2c.

Figure 3c

Figure 3f

Supplementary Figure 9

Figure 2c

7. Finally, I am not too convinced of the model that the authors propose: The binding affinities of masitinib and imatinib to dCK are low micromolar and their binding site is the nucleoside binding pocket. Hence, masitinib and imatinib should rather act as competitive inhibitors than activators. But I guess that based on the data that is available, it will be difficult to really verify the proposed model. For this, kinetic data, including k_{on} and k_{off} rates for both substrates and TKIs would be necessary and the residence times would need to be measured. Without this, the model that masitinib promotes product release remains a pure speculation that is not supported by data. Revise and tone-down claims.

We understand these remarks from Reviewer #1. We think that the reason why masitinib and imatinib are activators rather than inhibitors could be a small residence time for these molecules due to a fast binding kinetics. We agree with Reviewer #2 on the fact that to verify the proposed model, the K_{on} and K_{off} rates for both substrates and TKIs should be determined and we have made a huge effort in this direction.

To measure the K_{on} and K_{off} for TKIs and substrates, we used a Biolayer Interferometry (BLI) binding assay on an Octet Red96 instrument. dCK was biotinylated in a 1:1 ratio (one biotin per protein ratio confirmed by HABA assay) and we verified by Thermal Shift Assay (TSA) that the bindings affinities for TKIs and substrates were similar before and after biotinylation. Biotinylated dCK was immobilized upon Super Streptavidin Sensors and binding experiments in the presence of TKIs (masitinib, imatinib) and substrates (2'dC, gemcitabine) were performed. Data were processed and K_D , K_{on} and K_{off} values were determined using the global fitting procedures as implemented in the ForteBio Data Analysis Software.

Unfortunately, we could not detect any signal for substrates binding in BLI assays even though we validated substrates binding to biotinylated dCK by an orthogonal binding assay (TSA). BLI read out is a direct measure of changes in thickness of the biological layer which is probably incompatible with the detection of small compact molecules (MW ~ 200) in the buried substrate cavity of dCK.

Nevertheless, we were able to detect the binding of the larger TKIs inhibitors in the open form of immobilized dCK and measure kinetics parameters (Supplementary Figure 12 and Supplementary Table 4). In these experiments, K_D values for masitinib and imatinib (respectively 1.2 μ M and 3.3 μ M) were similar to previously determined data by ITC, confirming the low micromolar binding range affinity. For both compounds, binding was characterized by a very fast off rate ($k_{off} \sim 10^{-1} \text{ s}^{-1}$), with residence times ranging from 2 (imatinib) to 5 seconds (masitinib), reflecting the very fast and transient nature of the complex with dCK. In order to strengthen our model hypothesis we sought to compare this kinetics behaviour with a known dCK inhibitor that binds in the same pocket. We synthesized and measured the binding kinetics of a known dCK inhibitor (DI-39)^{7,8}. Although the signal observed for this compound was close to the technical detection limit of the method, we were able to reproducibly measure the binding to dCK and determine the kinetics parameters of this interaction. We measured a slower off rate ($k_{off} \sim 10^{-2} \text{ s}^{-1}$), with a higher residence time ($t_R = 23 \text{ s}$), showing a significant difference between activators (masitinib, imatinib) and an inhibitor (DI-39). Consequently, this very fast off rate supports our proposed model, where TKIs are capable to bind dCK and promote the release of the reaction product due to their high affinity but are not competitive inhibitors because of their low residence time and the lack of strong polar interaction with dCK, as shown in the crystal structures.

Moreover, we think that, rather than off rate of substrates, the most important data to validate the model would be the off rates for the enzymatic products release (monophosphate substrates) to be compared with the values measured for the TKIs. However, this kind of experiments would require enzymatic assays using radioligands and will be very difficult to set up. For this reason we have decided, in agreement with Reviewer #1 comment, to tone down our conclusions in the discussion by clarifying that our working model is still a working hypothesis that requires more experiment to be validated (lines 341-344).

	ITC	BLI			
	K_D (μ M)	K_D (μ M)	k_{on} (1/Ms)	k_{off} (1/s)	t_R (s)
2'dC	2.8 \pm 0.1	-	-	-	-
Gemcitabine	22.2 \pm 1.1	-	-	-	-
Masitinib	1.4 \pm 0.1	1.6 \pm 0.4	1.2 \pm 0.6 $\times 10^5$	1.8 \pm 0.4 $\times 10^{-1}$	5.7 \pm 1.2
Imatinib	2.0 \pm 0.8	4.7 \pm 2.3	1.2 \pm 0.2 $\times 10^5$	3.9 \pm 0.2 $\times 10^{-1}$	2.6 \pm 0.1
DI-39	-	0.54 \pm 0.28	3.4 \pm 2.7 $\times 10^5$	0.45 \pm 0.07 $\times 10^{-1}$	23.0 \pm 3.8

Supplementary Figure 12

Reviewer #2 (Remarks to the Author):

In the work by Hammam et al an interesting mechanism is discovered potentially explaining synergies between masitinib and gemcitabine. Using affinity proteomics they discover an interaction between deoxycytidine kinase (dCK) and masitinib that is confirmed by biophysical and structural studies. Although masitinib binding overlap with the substrate nucleoside binding site they propose that the ligand serve as an activator of dCK, rather than an inhibitor. Using activity studies of a subset of nucleoside substrates, including drugs, they provide support for that this might constitute a general activation mechanism of dCK. Furthermore, they show that there is a correlation between dCK expression and potentially synergistic effects in cell lines.

Overall the work is quite well presented and the combined data supports the conclusions that this mechanism is contributing to the synergies seen for the two drugs in cell lines. I think this discovery at the cell line level, which also has implications for the development of dCK activating drugs, is enough to merit the work for publication in Nature Communications, in spite of that other mechanism might be operative for the synergies seen in some clinical studies.

Still, there are quantitative aspects of the effect of this activation mechanism which would be interesting to shed further light on and, for example, measurement of the changes in intracellular levels of the active adduct of gemcitabine would strengthen the work very significantly (e.g. see methods from Per Artursson's group).

As Reviewer #2 suggested, we have investigated the intracellular levels of the active adduct of gemcitabine. However we think that the proposed methods from Per Artursson's group are not suitable for the determination of the intracellular gemcitabine metabolites (gemcitabine-monophosphate, gemcitabine-diphosphate and gemcitabine-triphosphate). Since these metabolites are negatively charged as nucleotides, the most used determination method is HPLC (ion exchange, ion pair or porous graphitic carbon)⁹. We tried, as an alternative to the suggested methods, a Cellular Nucleotide Profiling Analytical Service provided by NovoCIB, a French start-up specialized in analytical quantification of nucleotides by HPLC.

To study the effect of nucleoside analogues on the whole spectra of cellular purine, pyrimidine, deoxy- and ribonucleotides, they have developed an original cell-based analytical approach in which (deoxy)ribonucleotides (mono-, di-, triphosphate) and nucleotide co-factors are extracted from cultured cells, separated by ion-paired chromatography and quantified. These cellular assays have been validated with antiviral and anti-cancer nucleoside analogues.

As the gemcitabine-monophosphate is the unique adduct directly phosphorylated by dCK, we tried to determine its concentration in cellular samples treated with gemcitabine and/or masitinib. Unfortunately, we

were not able to determine gemcitabine monophosphate in cellular samples. Even if the small signal for gemcitabine monophosphate was well separated and identified in standards samples, cellular extracts presented a large signal in the same region which prevented its identification and quantification.

I also have problems with one of the quantifications in figure 1. In fig 1c it is shown the activation of gemcitabine (at inhibitor/protein ratio 22) is 3 times stronger than for dC. However, if you compare gemcitabine at an inhibitor/protein ratio of 22 in fig1b it is only 1.5 times stronger. For the mechanism this makes little different but why is the data not consistent?

We thank Reviewer #2 for noticing this inconsistency. Obviously, we made a mistake when plotting the data for 2'dC in Figure 2c. The raw data, the processed data and the conclusions drawn were not affected so the discussion remains the same in the manuscript. The figure 2c has been corrected. We are really sorry and we apologize for this mistake.

Figure 2c

Concerning the somewhat controversial activation mechanism where the binding is overlapping with the substrate-binding site, and presumably assisting in releasing the product/opening the active site, the authors should take the discussion a bit further. For example, there are probably other enzymes with an activation mechanisms which can serve as a precedent for this type of scenario, and they should find references for those.

We have extended the discussion and compared our proposed activation model with other activation mechanisms described elsewhere.

“There are very few examples of mechanistically well characterized small-molecule enzyme activators. These activators include regulators of proteases, kinases, deacetylases, dehydrogenases, phosphatases and nucleases¹⁰. Four main types of mechanism are listed: the binding of a small molecule to an allosteric site directly in the catalytic domain to promote an active conformation (type A1) (PDK1¹¹, GK)¹²; the binding to an allosteric site to facilitate an irreversible-activating post-translational modification (type A2) (Procaspase-3)¹³; the binding to a regulatory subunit to indirectly promote activity at the catalytic domain (type B1) (AMPK)¹⁴; and binding to a regulatory subunit to promote an activating oligomerization (type B2) (RNaseL)¹⁵. Recent examples that illustrate other mechanisms could also lead to small-molecule activation, but they are described as unpredictable and difficult to understand. For aldehyde dehydrogenase 2 (ALDH2), the small molecule

activator partially blocks the substrate entrance tunnel, but the substrate access and product release can still occur¹⁶. Instead of interfering with the enzymatic activity, it leaves the active site residues free to function and increases the enzymatic rate. The authors propose that by binding and blocking one of the active site exits, the activator increases the likelihood of a productive encounter between partners, improving catalytic efficiency. Among the few other examples, we can cite sirtuin-1 (SIRT1), for which activation with small compounds has been observed but the mechanism remains unclear and is the subject of intense controversy because two opposing models were proposed¹⁷.

dCK activation could be a particular scenario of type A1 activation, as the binding of TKIs in the catalytic domain promote a specific conformation, which increases enzymatic activity. However, the binding site is not purely allosteric, as the bottom of the cavity partially overlaps with the substrate binding. The location of TKIs binding within the substrate binding site of dCK is similar to the binding of a few other molecules such as the F-series (F1, F2, F3, F4)¹⁸ and the DI-39⁸ compound, known as potent inhibitors of dCK. The opposite effect of TKIs (activators) and these compounds (inhibitors) on dCK can be partially explained when analysing their crystal structures contact with dCK. Indeed, in the F-inhibitor (or DI-39) bound structure, the catalytic carboxylic acid of the residue Glu53 (catalytic residue responsible for activating the 5'-hydroxyl group of the nucleoside for nucleophilic attack on UTP), makes a 3.2 (or 2) Å hydrogen bond interaction with one of the two exocyclic amine groups^{8,18}. These compounds interact directly with an essential active site residue, inhibiting the enzyme by restricting substrate binding and catalysis. In contrast, our structures show that TKIs binds partially at the active site, but without strong direct hydrogen bond interactions and without sterically interfering with the catalytic residues. Because TKIs only partially block the substrate site with smaller residence time than inhibitors, we suggest that TKI do not interfere with substrates binding and promote an activation of the enzyme activity rather than an inhibition as observed for stronger binders. The very fast off rate observed for TKIs supports our proposed model, where TKIs are capable to bind dCK and promote the release of the reaction product due to their high affinity but they are not competitive inhibitors because of their low residence time and the lack of strong polar interaction with dCK, as shown in the crystal structures. These activators bind the phospho-nucleoside-bound form of dCK-C and promote the transition from the 'closed' to the 'open' conformation, critical for the release of enzymatic products. This conformational change accelerates the rate-limiting step of the reaction independently of UTP, which permits dCK to bind to the phosphate donor and to continue its enzymatic cycle."

Overall, it looks like the manuscript needs a bit more work and there are also a number of statements in the paper referring to literature data but where the reference is missing.

We apologize to the reviewer but we could not find the mentioned missing references in the manuscript. We will be very grateful if Reviewer #2 could provide us with more details about this particular point and we will be very happy to correct it accordingly.

References

1. Di Veroli, G.Y. et al. Combenefit: an interactive platform for the analysis and visualization of drug combinations. *Bioinformatics* **32**, 2866-8 (2016).
2. Anastassiadis, T., Deacon, S.W., Devarajan, K., Ma, H. & Peterson, J.R. Comprehensive assay of kinase catalytic activity reveals features of kinase inhibitor selectivity. *Nat Biotechnol* **29**, 1039-45 (2011).
3. Rix, U. et al. Chemical proteomic profiles of the BCR-ABL inhibitors imatinib, nilotinib, and dasatinib reveal novel kinase and nonkinase targets. *Blood* **110**, 4055-63 (2007).
4. Meuth, M. Deoxycytidine kinase-deficient mutants of Chinese hamster ovary cells are hypersensitive to DNA alkylating agents. *Mutat Res* **110**, 383-91 (1983).
5. Johnson, A.J., Brown, M.N. & Black, M.E. Evaluation of a UCMK/dCK fusion enzyme for gemcitabine-mediated cytotoxicity. *Biochem Biophys Res Commun* **416**, 199-204 (2011).
6. Jordheim, L.P., Galmarini, C.M. & Dumontet, C. Gemcitabine resistance due to deoxycytidine kinase deficiency can be reverted by fruitfly deoxynucleoside kinase, DmdNK, in human uterine sarcoma cells. *Cancer Chemother Pharmacol* **58**, 547-54 (2006).
7. Murphy, J.M. et al. Development of new deoxycytidine kinase inhibitors and noninvasive in vivo evaluation using positron emission tomography. *J Med Chem* **56**, 6696-708 (2013).
8. Nathanson, D.A. et al. Co-targeting of convergent nucleotide biosynthetic pathways for leukemia eradication. *J Exp Med* **211**, 473-86 (2014).

9. Micova, K., Friedecky, D. & Adam, T. Mass Spectrometry for the Sensitive Analysis of Intracellular Nucleotides and Analogues. in *Mass Spectrometry* (ed. Aliofkhazraei, M.) (InTech, 2017).
10. Zorn, J.A. & Wells, J.A. Turning enzymes ON with small molecules. *Nat Chem Biol* **6**, 179-188 (2010).
11. Engel, M. et al. Allosteric activation of the protein kinase PDK1 with low molecular weight compounds. *EMBO J* **25**, 5469-80 (2006).
12. Grimsby, J. et al. Allosteric activators of glucokinase: potential role in diabetes therapy. *Science* **301**, 370-3 (2003).
13. Wolan, D.W., Zorn, J.A., Gray, D.C. & Wells, J.A. Small-molecule activators of a proenzyme. *Science* **326**, 853-8 (2009).
14. Xiao, B. et al. Structural basis of AMPK regulation by small molecule activators. *Nat Commun* **4**, 3017 (2013).
15. Thakur, C.S. et al. Small-molecule activators of RNase L with broad-spectrum antiviral activity. *Proc Natl Acad Sci U S A* **104**, 9585-90 (2007).
16. Perez-Miller, S. et al. Alda-1 is an agonist and chemical chaperone for the common human aldehyde dehydrogenase 2 variant. *Nat Struct Mol Biol* **17**, 159-64 (2010).
17. Hubbard, B.P. & Sinclair, D.A. Small molecule SIRT1 activators for the treatment of aging and age-related diseases. *Trends Pharmacol Sci* **35**, 146-54 (2014).
18. Nomme, J. et al. Structural characterization of new deoxycytidine kinase inhibitors rationalizes the affinity-determining moieties of the molecules. *Acta Crystallogr D Biol Crystallogr* **70**, 68-78 (2014).

Reviewers' Comments:

Reviewer #1 (Remarks to the Author):

The authors have addressed all points raised during the initial round of review to my satisfaction.
No further comments.